# A designer FG-Nup that reconstitutes the selective transport barrier of the nuclear pore complex

Alessio Fragasso [1,4], Hendrik W. de Vries [2,4], John Andersson [3], Eli O. van der Sluis[1],
Erik van der Giessen [2], Andreas Dahlin[3], Patrick R. Onck [2✉] & Cees Dekker [1✉]

Nuclear Pore Complexes (NPCs) regulate bidirectional transport between the nucleus and the cytoplasm. Intrinsically disordered FG-Nups line the NPC lumen and form a selective barrier, where transport of most proteins is inhibited whereas specific transporter proteins freely pass. The mechanism underlying selective transport through the NPC is still debated. Here, we reconstitute the selective behaviour of the NPC bottom-up by introducing a rationally designed artificial FG-Nup that mimics natural Nups. Using QCM-D, we measure selective binding of the artificial FG-Nup brushes to the transport receptor Kap95 over cytosolic proteins such as BSA. Solid-state nanopores with the artificial FG-Nups lining their inner walls support fast translocation of Kap95 while blocking BSA, thus demonstrating selectivity. Coarse-grained molecular dynamics simulations highlight the formation of a selective meshwork with densities comparable to native NPCs. Our findings show that simple design rules can recapitulate the selective behaviour of native FG-Nups and demonstrate that no specific spacer sequence nor a spatial segregation of different FG-motif types are needed to create selective NPCs.

[1] Department of Bionanoscience, Kavli Institute of Nanoscience, Delft University of Technology, Delft, The Netherlands. [2] Zernike Institute for Advanced Materials, University of Groningen, Groningen, The Netherlands. [3] Department of Chemistry and Chemical Engineering, Chalmers University of Technology, Gothenburg, Sweden. [4]These authors contributed equally: Alessio Fragasso, Hendrik W. de Vries. ✉email: p.r.onck@rug.nl; c.dekker@tudelft.nl

Nucleocytoplasmic transport is orchestrated by the nuclear pore complex (NPC), which imparts a selective barrier to biomolecules[1,2]. The NPC is a large eightfold symmetric protein complex (with a size of ~52 MDa in yeast and ~112 MDa in vertebrates) that is embedded within the nuclear envelope and comprises ~30 different types of Nucleoporins (Nups)[3,4]. Intrinsically disordered proteins, termed phenylalanine-glycine (FG)-Nups, line the central channel of the NPC. FG-Nups are characterized by the presence of FG repeats separated by spacer sequences[5] and they are highly conserved throughout species[6]. FG-Nups carry out a dual function: by forming a dense barrier (100–200 mg/mL) within the NPC lumen, they allow passage of molecules in a selective manner[7–10]. Small molecules can freely diffuse through, whereas larger particles are generally excluded[11]. At the same time, FG-Nups mediate the transport of large nuclear transport receptor (NTR)-bound cargoes across the NPC through transient hydrophobic interactions between FG repeats and hydrophobic pockets on the convex side of NTRs[12]. Various models have been developed in order to connect the physical properties of FG-Nups to the selective properties of the NPC central channel, e.g., the 'virtual-gate'[13], 'selective phase'[14,15], 'reduction of dimensionality'[16], 'kap-centric'[17–19], 'polymer brush'[20] and 'forest'[5] models. No consensus on the NPC transport mechanisms has yet been reached.

The NPC is highly complex in its architecture and dynamics, being constituted by many different Nups that simultaneously interact with multiple transiting cargoes and NTRs. In fact, the NTRs with their cargoes may amount to almost half of the mass of the central channel, so they may be considered an intrinsic part of the NPC[3]. These NPC properties complicate in vivo studies[3,21–23], for which it is very challenging to identify contributions coming from individual FG-Nups[24,25]. On the other hand, in vitro approaches to study nucleocytoplasmic transport using biomimetic NPC systems[26–33] have thus far been limited to single native FG-Nups and mutations thereof, attempting to understand the physical behaviour of FG-Nups and their interactions with NTRs. The reliance on a few selected Nups from yeast or humans in these studies with sequences that evolved over time in different ways for each of these specific organisms makes it difficult to pinpoint the essential and minimal properties that provide FG-Nups with their specific selective functionality.

Here, we describe a bottom-up approach to studying nuclear transport selectivity, where we rationally design, synthesize, and assess artificial FG-Nups with user-defined properties that are set by an amino acid (AA) sequence that is chosen by the user. With this approach we address the question: can we build a synthetic protein that mimics the selective behaviour of native FG-Nups? By combining experiments and coarse-grained molecular dynamics (MD) simulations, we illustrate the design and synthesis of an artificial 311-residue long FG-Nup, which we coin NupX, and characterize its selective behaviour with respect to Kap95 (a well-characterized NTR from yeast, 95 kDa) versus bovine serum albumin (BSA, 66 kDa). First, we explore the interactions between Kap95 and NupX brushes with varying grafting densities using quartz crystal microbalance with dissipation monitoring (QCM-D), finding that NupX brushes bind Kap95 while showing no binding to BSA. We confirm this finding by calculating the potential of mean force (PMF) associated with the entry of Kap95 or an inert cargo into NupX brushes. Second, we explore the transport properties of NupX-functionalized solid-state nanopores and show that NupX-lined pores constitute a selective transport barrier. Similar to FG-Nups previously studied with the same technique[28,31], the NPC-mimicking nanopores allow fast and efficient passage of Kap95 molecules, while blocking transport of BSA. Coarse-grained MD simulations of NupX-functionalized nanopores highlight the formation of a dense FG-rich meshwork with similar protein densities as in native NPCs, which excludes inert molecules but allows entry and passage of Kap95.

The current work provides the proof of concept that a designer FG-Nup can reconstitute NPC-like selectivity, and the results show that no specific spacer sequence nor a spatial segregation of different FG-motifs (as observed in recent work[3,34]) is required for achieving selectivity. This work lays the foundation for multiple future directions in follow-up work as the approach opens the route to systematically study the essential microscopic motifs that underlie the unique selectivity of NPCs.

## Results

**Design of the synthetic NupX.** In the design of our synthetic NupX protein, we aim to reconstitute nuclear transport selectivity while operating under a minimal set of simple design rules. The design procedure that we outline below uses the following four rules: (i) we design a protein that incorporates the physical properties of GLFG-Nups (a specific class of essential FG-Nups that are particularly cohesive and contain many glycine-leucine-phenylalanine-glycine (GLFG)-motifs), (ii) it comprises two parts, with a cohesive domain at one end and a repulsive domain at the other end, where each domain is characterized by the ratio C/H of the number of charged and the number of hydrophobic residues, (iii) FG- and GLFG-motifs are present in an alternating and uniformly spaced fashion within the protein's cohesive domain and (iv) the protein is intrinsically disordered throughout its full length, similar to native FG-Nups.

We implemented our design rules in a stepwise design process as follows. First, we selected and analyzed an appropriate set of native FG-Nups (design rule i), namely GLFG-Nups, which differ from other FG-Nups in terms of the type of FG repeats and the properties of the spacer regions[11]. The emphasis on GLFG-Nups follows from their localization in the central channel[3] of the yeast NPC (Fig. 1a), where they strongly contribute to the nuclear transport selectivity. Indeed, a small subset of GLFG-Nups (e.g., either Nup100 or Nup116 in combination with Nup145N) was shown to be essential and sufficient for cell viability[21,35]. To derive the AA content of NupX, we therefore characterized the archetypical GLFG-Nup sequence by determining the AA content of the disordered regions of Nup49, Nup57, Nup145N, Nup116 and Nup100 from yeast. Of these, the most essential GLFG-Nups (i.e., Nup100, Nup116 and Nup145N) comprise a collapsed domain with a low C/H-ratio and abundance of FG/GLFG repeats, and an extended domain with a high C/H-ratio and absence of FG repeats[5]. This distinction is highlighted in Fig. 1b, c, where non-FG/GLFG/charged residues are highlighted in light green and pink for the collapsed and extended domains, respectively—a colouring scheme used throughout this work. The division into two domains of these essential GLFG-Nups led us to phrase design rule ii in our design process of NupX, with each domain comprising ~150 AA residues (see Fig. 1b, c), whereas the extended domain of NupX is of quite similar length to the corresponding extended domains of Nup100, Nup116 and Nup145N (190 residues on average), the cohesive domain is notably shorter than the collapsed domains of native GLFG-Nups (390 residues on average).

Assigning the AA content to NupX, as derived from the sequence information of the GLFG-Nups, was performed separately for the two domains: we computed the cumulative AA contents (excluding FG- and GLFG-motifs) for both the collapsed domains of all five GLFG-Nups, and for the extended domains of Nup100, Nup116 and Nup145N (design rule ii). Upon normalizing for the total length of the collapsed or extended domains of all native GLFG-Nups, this analysis resulted

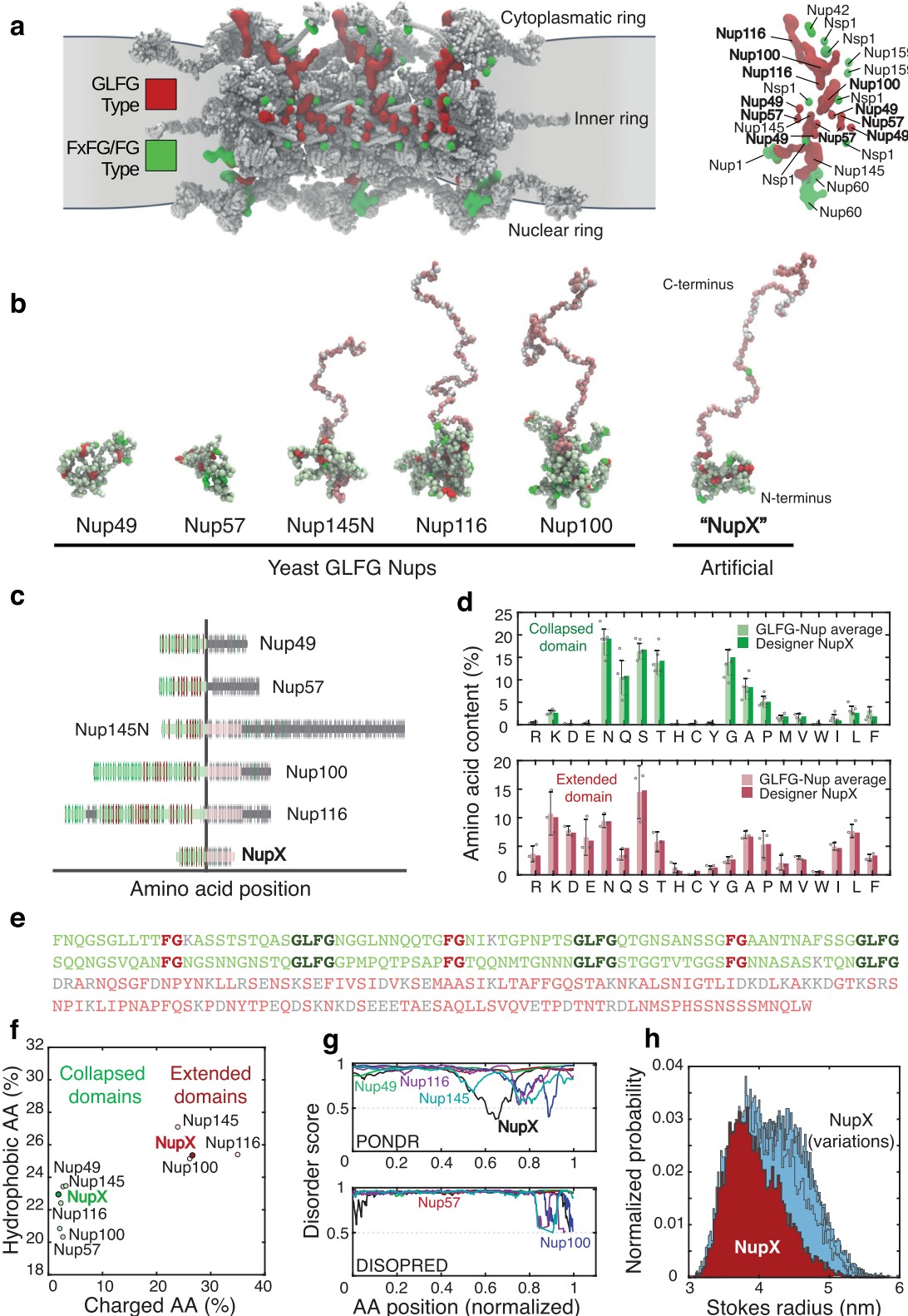

in the distributions presented in Fig. 1d, plotted separately for the collapsed (light green, top) and the extended (light red, bottom) domains. Based on these histograms, we assigned AAs to the collapsed and extended domains of NupX separately. Following design rule iii, we then placed FG- and GLFG repeats in the collapsed domain with a fixed spacer length of ten AAs. This

value was chosen based on the spacer length of ~5–15 AAs in native GLFG-Nups. An analysis of the charged and hydrophobic AA content of the domains of NupX and native GLFG-Nups shows that the assigned sequence properties are indeed reproduced by our design method (Fig. 1f). Finally, the sequences of the collapsed and extended domains of NupX were repetitively

**Fig. 1 De novo design of an artificial FG-Nup. a** Left: frontal view on three spokes of the *Saccharomyces cerevisiae* NPC (PDB-DEV, entries 11 and 12, Ref. [3]) that shows how the GLFG-Nups (red) are predominantly anchored in the inner ring, as opposed to the FxFG/FG-Nups (green). Right: anchoring points of individual Nups in a single spoke. The GLFG-Nups Nup100, Nup116, Nup49 and Nup57 (red) contribute strongly to the permeability barrier of the NPC[3], where Nup100 and Nup116 are known to be indispensable for NPC viability[21,35]. This image and other visualizations of protein structures were rendered using VMD[80]. **b** Simulation snapshots of isolated native yeast GLFG-Nups at one amino acid resolution. The conformations of Nup145N, Nup116 and Nup100 highlight a bimodality of the Nups[5], with a collapsed and extended domain. FG repeats, GLFG repeats, and charged residues are displayed in bright green, red, and white, respectively. Other amino acids in the cohesive and extended domains are depicted in light green and pink, respectively. NupX adopts the same bimodal conformations as essential GLFG-Nups Nup100 and Nup116. **c** Comparison of the full-length sequences between yeast GLFG-Nups and NupX. Sequence highlights follow the colour scheme of panel **b**, folded domains are indicated in dark-grey. **d** Amino acid contents of yeast GLFG-Nups (averaged) and NupX for the collapsed (top panel) and extended (bottom panel) domains. Bar heights denote the average amino acid fraction within GLFG-Nup domains, where $N = 5$ (all GLFG-Nups) for the collapsed domain and $N = 3$ (Nup100, Nup116, Nup145) for the extended domain. Error bars indicate standard deviations in the average occurrence of amino acids. FG- and GLFG-motifs were excluded from this analysis. **e** Sequence of NupX, following the colour scheme of **b** and **c**. FG and GLFG repeats are spaced by ten residues in the cohesive domain. **f** Charge-and-hydrophobicity plot of NupX and yeast GLFG-Nup domains. For both the collapsed (green shading) and extended (red shading) domains, the charged and hydrophobic amino acid contents of NupX agree with the properties of individual GLFG-Nups. **g** Disorder prediction scores for the unfolded domains of GLFG-Nups (coloured lines) and full-length NupX (black curve) from two different predictors (see Methods). Disorder prediction scores higher than 0.5 (dashed line) count as fully disordered. **h** Distribution of Stokes radii from 10 μs of coarse-grained molecular dynamics simulations for NupX (red) and 25 design variations (light blue). NupX is, on average, slightly more compacted than other design variants.

shuffled (except for the FG- and GLFG-motifs that we kept fixed) until a desirable level of disorder was achieved (design rule iv), as predicted by PONDR[36] and DISOPRED[37,38] (Fig. 1g). This resulted in the NupX sequence shown in Fig. 1e. Whereas PONDR predicts one short folded segment between residues 189 and 209 (normalized position of 0.65 in Fig. 1g), additional structure prediction[39] (Methods) did not yield any high-confidence folded structures for this segment.

To assess the robustness of our design procedure, we tested how permutations of the NupX sequence (which shuffle AAs while retaining the FG/GLFG sequences and the definition of both domains) affect the Stokes radius $R_s$, as computed from one-bead-per-amino-acid (1BPA) MD simulations developed for intrinsically disordered proteins (Fig. 1h, see Methods). We found that 25 different designs for NupX (Supplementary Table 2) yielded an average $R_s$ of $4.2 \pm 0.2$ nm (errors are SD). This is close to the simulated ($3.9 \pm 0.4$ nm) and measured ($3.7 \pm 1.1$ nm by dynamic light scattering (DLS), Supplementary Table 1) $R_s$ value of the NupX protein design (Fig. 1e).

Summing up, using a minimal set of rules, we designed a NupX protein that incorporates the average properties that characterize GLFG-Nups[5,11]. Moreover, by creating 25 different designs that all showed similar behaviour in our simulations, we showed that the physical properties such as the Stokes radius and the division of NupX into a cohesive and repulsive domain are recovered in a reliable way.

**QCM-D experiments and MD simulations show selective binding of Kap95 to NupX brushes.** To assess the interaction between NupX and Kap95, we employed a QCM-D, with gold-coated quartz chips and phosphate-buffered saline (PBS, pH 7.4) as running buffer, unless stated otherwise. First, C-terminus-thiolated NupX molecules were injected into the chamber at a constant flow-rate (20 μL/min) where they chemically reacted with the gold surface. Binding of NupX to the gold surface could be monitored in real time by measuring the shift in resonance frequency Δf of the quartz chip (Fig. 2a). We applied the NupX coating by administering a protein concentration ranging from 100 nM to 2 μM (Supplementary Fig. 2) until a plateau in the frequency shift was reached, which typically occurred after ~1 h of incubation. To gain insight into the areal mass density of the deposited layers, we employed surface plasmon resonance (SPR) measurements (Supplementary Fig. 3), where we used the same coating protocol for consistency. From these measurements of the

areal mass density, we found grafting distances of $7.7 \pm 0.5$ nm (mean $\pm$ SD) for chips incubated with a 60-nM NupX solution, and $2.91 \pm 0.02$ nm (mean $\pm$ SD) for 2 μM. In determining the grafting density from the areal mass density, we assumed a tri-angular lattice (since an equilateral triangulated (sometimes also denoted as hexagonal) lattice is the densest type of packing that can be described by a unique length scale that sets the grafting density). Figure 2a shows a typical frequency shift over time for the binding of 1 μM NupX to a gold surface. After the Nup-layer was formed, a 1-mercapto-11-undecylte-tri(ethyleneglycol) molecule (MUTEG), which is expected to form a ~2-nm thin passivating film[17], was added to passivate any remaining bare gold that was exposed in between NupX molecules (Supplementary Fig. 4). This minimizes unintentional interactions between Kap95 and gold for subsequent binding experiments (Supplementary Fig. 5)[17,18,40].

Thus, after setting up a NupX-coated layer, we flushed in Kap95 at stepwise increasing concentrations (~10–3000 nM, Fig. 2d) and monitored binding to the NupX-coated surface. We observed a clear concentration-dependent amount of Kap95 molecules bound to the NupX brush. For reference, we repeated the experiment on brushes of Nsp1 (a native FG-Nup from yeast), as well as Nsp1-S, a Nsp1-mutant where the hydrophobic AAs F, I, L, V are replaced by the hydrophilic AA Serine (S) (Fig. 2b, c). The latter was employed as a negative control since it is expected to not bind Kap95 due to the lack of FG repeats[14,15]. Gold surfaces coated with Nsp1 or Nsp1-S were characterized with SPR under similar coating conditions as for QCM-D, yielding grafting distances of $4.9 \pm 0.1$ nm for Nsp1 and $5.8 \pm 0.4$ nm for Nsp1-S. Upon flushing Kap95, we found, consistent with previous studies[27,40], a concentration-dependent adsorption to Nsp1 brushes (Fig. 2e), whereas we did not observe any detectable interaction between Kap95 and Nsp1-S (Fig. 2f). The latter is consistent with the lack of FG repeats in the Nsp1-S sequence that makes the Nsp1-S film devoid of binding sites for Kap95. We note that non-linear effects, e.g., coverage-dependent changes in water entrapment within the layer[41], are likely to affect the observed equilibrium signals, which, together with a relatively slow dissociation of Kap95 from both the NupX and Nsp1 brushes, led us to refrain from extracting a dissociation constant. Adsorbed molecules could be completely removed upon flushing 0.2 M NaOH however (Supplementary Fig. 6). Finally, we investigated whether the inert molecule BSA could bind to the NupX brush. Upon flushing 2.5 μM of BSA (Fig. 2g) we did not observe any appreciable change in the resonance frequency,

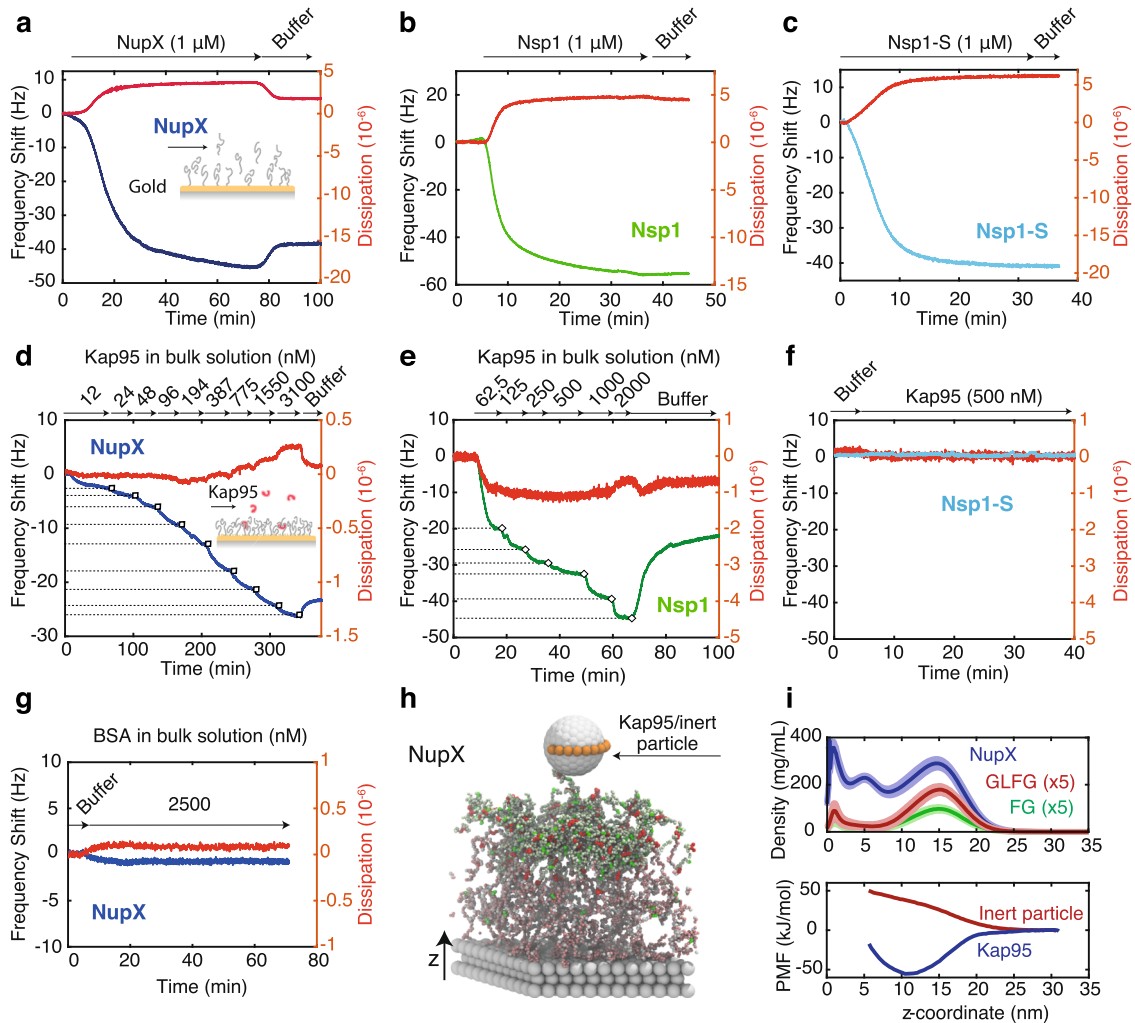

**Fig. 2 Binding affinity of Kap95 to NupX, Nsp1 and Nsp1-S brushes, using QCM-D and MD simulations. a–c** Change in frequency shift upon coating of gold surface with NupX (dark blue), Nsp1 (green) and Nsp1-S (light blue) proteins, respectively, at 1 µM protein concentration. Red curves indicate the corresponding shift in dissipation. **d–f** Change in frequency shift upon titration of Kap95 (with concentration in the range ~10–3000 nM) on NupX (dark blue), Nsp1 (green) and Nsp1-S (light blue) coated surfaces. Numbers indicate the concentration in nM of Kap95 for each titration step. Large changes in frequency shift are observed for NupX and Nsp1, whereas no detectable shift is measured for Nsp1-S. Red curves indicate the corresponding shift in dissipation. **g** Frequency (dark blue) and dissipation (red) shift upon adsorption of 2.5 µM BSA onto the NupX-coated sensor. **h** Side-view snapshot of the umbrella sampling simulation setup for a NupX brush with 4.0 nm grafting distance, where a model Kap95 particle (8.5 nm diameter grey sphere with binding sites depicted in brown) or inert particle (7.5 nm diameter, not shown) is restrained along different z-coordinates. Scaffold beads are shown in grey, NupX proteins follow the same colour scheme as presented in Fig. 1b. **i** Top panel: time and laterally averaged protein density distributions for the NupX brushes (blue) and for the FG-motifs (green) and GLFG-motifs (red) present inside the NupX brushes with a grafting distance of 4.0 nm. The density profiles of the GLFG- and FG-motifs within the NupX brush are multiplied by 5 for clarity. Dark central lines and light shades indicate the mean and standard deviation in density profiles, respectively. These measures were obtained by averaging over the density profiles of trajectory windows 50 ns in length ($N = 60$). High-density regions (up to almost 400 mg/mL) form near the attachment sites ($z = 0$–2 nm) and near the free surface of the brush layer (at $z \sim 15$ nm). FG- and GLFG-motifs predominantly localize near the free surface of the brush. Bottom: free-energy profiles (PMF curves) of the centre of mass of the 8.5 nm sized model Kap95 (blue) and inert particle (red) along the z-coordinate, where $z = 0$ coincides with the substrate. The difference in sign between the PMF curves of both particles indicates a strong preferential adsorption of the model Kap95 to NupX brushes and a repulsive interaction with the inert particle.

indicating that the NupX brush efficiently excludes these inert molecules. This measurement was repeated for all the grafting conditions used in this study (Supplementary Fig. 7) showing that BSA did not produce any detectable shift in frequency, while Kap95 showed clear binding to the NupX films. Importantly, the data show that the NupX brush selectively interacts with Kap95 over a range of grafting densities.

In order to study the morphology and physical properties of NupX brushes at the microscopic level, we employed coarse-grained MD simulations (see Methods), which resolved the

density distribution within the NupX brush layer and the preferential adsorption of Kap95 over inert molecules of similar size such as BSA. Thirty-six NupX proteins were tethered onto a triangular lattice with a fixed spacing of 4.0 nm (Fig. 2h) or 5.7 nm (Supplementary Fig. 11), well in the range of grafting distances from 2.9 to 7.7 nm as measured by SPR. Averaged over a simulation time of 3 µs, we found that the NupX brushes with a 4.0 nm grafting distance form a laterally homogeneous mesh-work with densities ranging from ~400 mg/mL near the substrate to around ~200 mg/mL in the central region and to ~300 mg/mL

near the free surface of the brush (Fig. 2i, top panel). The interface near the free surface of the brush contains the highest relative concentration of FG- and GLFG-motifs (see Fig. 2i, top panel). Notably, the protein density throughout the brush is of the same order of magnitude as the density obtained in simulations of the yeast NPC[42]. Upon increase of the grafting distance to 5.7 nm, we find that the NupX brush attains different and less dense conformations: the density profile plateaus at a value of 170 mg/mL and slowly decays without showing a peak density near the free surface of the brush (Supplementary Fig. 11). We translated our density profiles into height estimates in a similar fashion as other computational studies on FG-Nup brushes[43,44]. We consider the $z$-coordinates at which 90% of the protein mass is incorporated as the effective brush heights. This approach yields brush heights of 12 and 18 nm for the NupX brushes with 5.7 and 4.0 nm grafting distances, respectively. These values coincide quite well with the inflection point of the decaying tail of the density profiles in Fig. 2l and Supplementary Fig. 11.

The simulated density profiles yield notably higher brushes than expected from the Sauerbrey equation; e.g., assuming a density of the hydrated brush of ~1.05 g/mL[45], one can estimate a brush height of 6.4 nm for the NupX brush in Fig. 2a (that was incubated at 1 μM, which we expect to have a grafting distance at the higher end of the values used in our simulations). Importantly, however, the Sauerbrey equation does not account for viscoelastic effects and only provides a lower limit to the brush height[41]. Indeed, given the dissipation-to-frequency ratio of ~$0.045 \times 10^{-6}$ Hz$^{-1}$ (Supplementary Fig. 9), one expects that the actual experimental brush height will be larger than 6.4 nm, an effect also seen in other QCM-D studies of FG-Nups[27,45]. Notably, a quantitative difference between the NupX brush height estimations of the computational and experimental results does not affect the major conclusions of the study, namely the selective transport across biomimetic nanopores with a rationally designed artificial FG-Nup and the selective binding of Kap95 over BSA to NupX.

To assess the selective properties of the NupX brushes, we performed umbrella sampling simulations of the adsorption of Kap95 and an inert molecule to NupX brushes (Methods), again for two grafting densities of 4 and 5.7 nm. We modelled Kap95 (Supplementary Fig. 12) as an 8.5 nm sized sterically repulsive (i.e., modelling only repulsive, excluded volume interactions) particle with ten hydrophobic binding sites[8,31,46,47] and a total net charge similar to that of Kap95 (−43e). The inert molecule was modelled as a sterically inert spherical particle of 7.5 nm diameter[10]. We obtained PMF curves associated with the adsorption of Kap95 and inert particles by means of the weighted histogram analysis method (WHAM)[48]. We found that for the dense brush (4.0 nm grafting density), a significant (−52 kJ/mol) negative free energy is associated with the entry of Kap95 in the NupX brush, as is visible in Fig. 2i (bottom panel). By contrast, the PMF curve of the inert particle steeply increased when the protein entered into the NupX meshwork, showing that adsorption of non-specific proteins of comparable size as Kap95 will not occur. The large free energy differences between Kap95 (corresponding to ~nM binding affinity) and inert particle adsorption qualitatively support the experimental findings. When increasing the grafting distance to 5.7 nm, the inert particle and Kap95 protein are repelled and adsorbed less strongly (μM binding affinity, similar to other in vitro works[49,50]), respectively (Supplementary Fig. 11). The data indicate that dense brushes bind more strongly to Kap95 and that selectivity is maintained for less densely coated NupX brushes, which is in line with our experimental observation of selective adsorption on QCM-D chips coated with NupX brushes of varying grafting densities (Supplementary Fig. 7).

## Single-molecule translocation experiments with NupX-coated nanopores demonstrate selectivity

In order to test whether our synthetic FG-Nup does indeed form a transport barrier that mimics the selective properties of the NPC, we performed electrophysiological experiments on biomimetic nanopores[28,31]. These NPC mimics were built by tethering NupX proteins to the inner walls of a solid-state SiN$_x$ nanopore[51] using self-assembled monolayer chemistry (details in Methods). Solid-state nanopores of 10–60 nm in diameter were fabricated onto a glass-supported[52] SiN$_x$ free-standing membrane by means of TEM drilling. A buffer with 150 mM KCl, 10 mM Tris, 1 mM EDTA, at pH 7.5 was used to measure the ionic conductance through the pores, while retaining near-physiological conditions. Coating bare SiN$_x$ pores with NupX yielded a significant decrease in conductance (e.g., ~50% for ~30 nm diameter SiN pores) of the bare-pore values, as estimated by measuring the through-pore ionic current before and after the functionalization (Supplementary Fig. 13). In addition, the current–voltage characteristic in the ±200 mV range (Supplementary Fig. 13) is linear both for the bare and NupX-coated pores, indicating that the NupX meshwork is not affected by the applied electric field at the 100 mV operating bias. To obtain more information on the NupX-coating process of our SiN$_x$ pores, we repeated the same functionalization procedure on silica-coated SPR chips (Supplementary Fig. 3), where APTES, Sulfo-SMCC and NupX coatings were independently characterized using the same coating protocol as for the SiN$_x$ nanopores, for consistency. From these experiments, we estimate an average grafting distance of 5.4 ± 1.1 nm between adjacent NupX molecules. Measurements of the ionic current through NupX-coated pores revealed a higher 1/f noise in the current (Supplementary Fig. 14) compared to bare pores, which we attribute to random conformational fluctuations of the Nups within the pore volume and access region[53,54], similar to findings from previous studies on biomimetic nanopores[28,31].

To test the selective behaviour of the biomimetic nanopore, we measured translocation rates of Kap95 and BSA through bare pores of ~30–35 nm in diameter (Fig. 3a). Figure 3c shows examples of raw traces recorded for a 30 nm pore under 100 mV applied bias, when either only buffer (top), 450 nM Kap95 (middle), or 2.8 μM BSA (bottom) was added to the cis-chamber. As expected, we observed transient dips in the current through the bare pore upon injection of the proteins, which we attribute to single-molecule translocations of the analyte molecules. As is typical in nanopore experiments, translocation events yield current blockades with a characteristic amplitude and dwell time, where the former relates to the size of the molecule occupying the pore and the latter generally depends on the specific interaction between the translocating molecule and the pore wall[55]. Next, we repeated the experiment under identical conditions on the same pore after coating with NupX took place (Fig. 3b). Examples of typical raw traces are shown in Fig. 3d. Strikingly, Kap95 molecules could still translocate efficiently through the NupX-coated pore, whereas BSA molecules were practically blocked from transport.

Figure 3e, f show scatter plots of the event distributions, where the conductance blockade is plotted against dwell time for all translocation events. For the bare pore, we observe similar average amplitudes of 0.24 ± 0.09 and 0.20 ± 0.05 nS (errors are SD) for BSA and Kap95, respectively. For the NupX-coated pore, we found slightly larger but again mutually similar event amplitudes of 0.31 ± 0.03 and 0.27 ± 0.03 nS for BSA and Kap95, respectively. We found comparable translocation times through the bare pore of 0.66 ± 0.03 and 0.81 ± 0.02 ms (errors are SEM) for BSA and Kap95, respectively. For the coated pore, however, we measured longer dwell times of 5.0 ± 0.5 and 1.9 ± 0.1 ms for BSA and Kap95, respectively, which indicates that the presence of the NupX

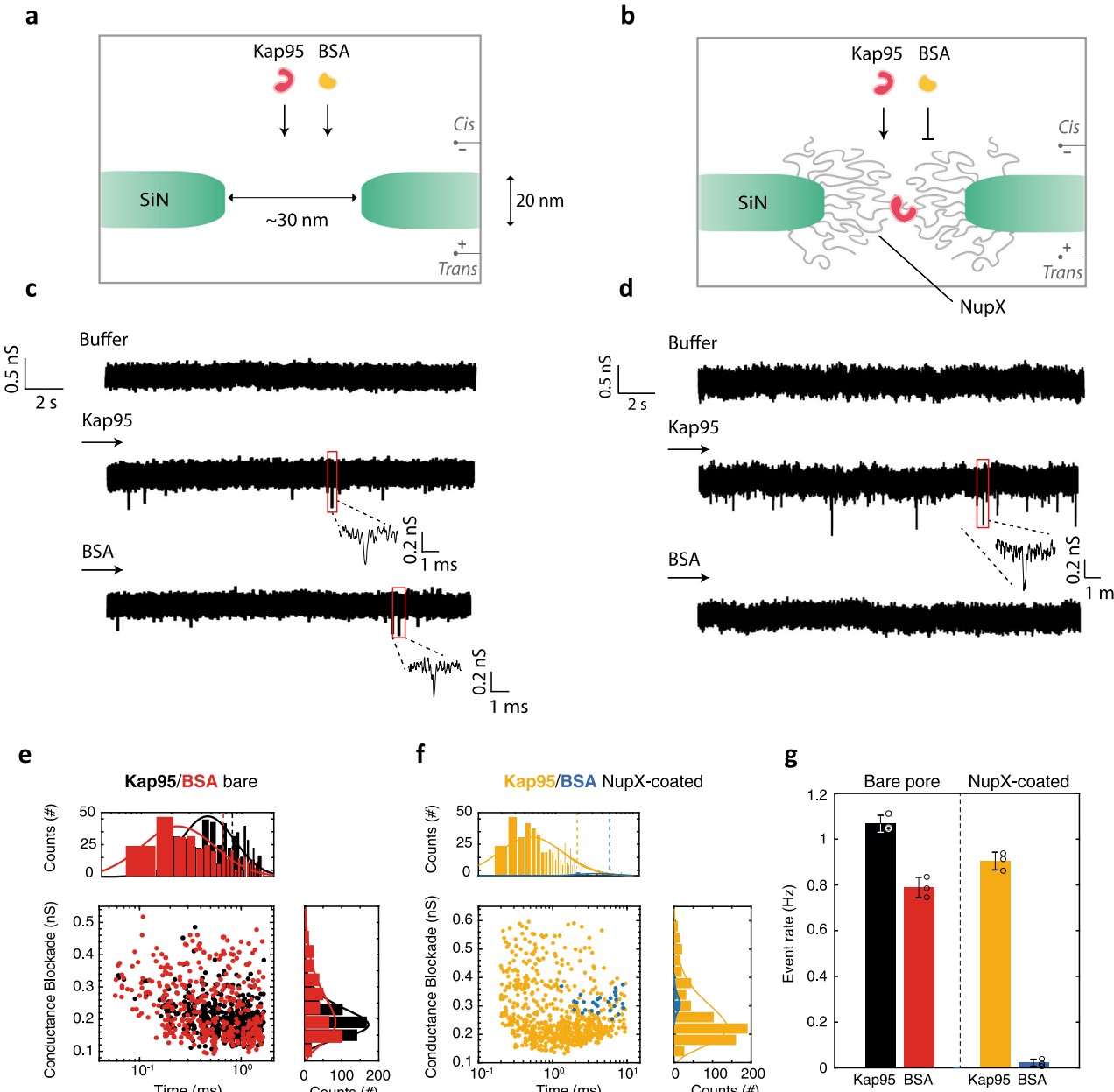

**Fig. 3 Electrical measurements on NupX-coated solid-state nanopores. a**, **b** Schematic of the nanopore system before (**a**) and after (**b**) NupX functionalization. **c, d** Examples of raw current traces through bare (**c**) and NupX-coated (**d**) pores, recorded under 100 mV applied bias for different analyte conditions. Current traces are recorded in the presence of buffer only (top), upon addition of 450 nM Kap95 (middle) and 2.8 μM BSA (bottom). Traces were filtered at 5 kHz. **e** Scatter plot showing conductance blockades and dwell time distributions of translocation events of the analytes Kap95 (black, $N = 506$) and BSA (red, $N = 387$) through a bare 30 nm pore, recorded over the same time interval. **f** Scatter plot showing conductance blockades and dwell time distributions of translocation events of the analytes Kap95 (yellow, $N = 686$) and BSA (blue, $N = 28$) through a NupX-coated 30 nm pore, recorded over the same time interval. Top and right panels in **e** and **f** show lognormal fits to the distribution of dwell times and conductance blockades, respectively. Dashed vertical lines in top panels indicate the mean values for the dwell time distributions. **g** Average event rate of translocations for Kap95 through a bare pore (black), BSA through a bare pore (red), Kap95 through a NupX-coated pore (yellow) and BSA through a NupX-coated pore (blue). Error bars indicate standard deviations from independent measurements (circles) on three different pores, $N = 3$.

molecules in the pore significantly slows down the translocation process of the passing molecules. Notably, BSA molecules were slower in translocating through the coated pore as compared to Kap95, which we attribute to the lower affinity between BSA and the NupX mesh as compared to Kap95. The transient and multivalent interactions between Kap95 and the FG repeats in the NupX meshwork lead to a reduced energy barrier as compared to BSA permeation, which may explain the observed differences in dwelling times[10]. Repeating the same experiment on a larger

60 nm NupX-coated pore (Supplementary Fig. 15) yielded selective pores with faster translocations for both Kap95 ($0.65 \pm 0.05$ ms) and BSA ($1.6 \pm 1.3$ ms), consistent with the presence of an open central channel. Smaller pores (<25 nm) did not result in any detectable signal for either Kap95 or BSA, due to the poor signal-to-noise ratio attainable at such low conductances.

Most importantly, these data clearly show selectivity of the biomimetic pores. Figure 3g compares the event rate of translocations for Kap95 and BSA through bare and NupX-

coated pores under 100 mV applied bias. Event rates were 0.79 ± 0.04 and 1.10 ± 0.04 Hz ($N = 3$ different nanopores; errors are SD) for BSA and Kap95 through the bare pore, respectively, whereas upon coating the pore with NupX, the event rates changed to 0.02 ± 0.02 and 0.90 ± 0.04 Hz ($N = 3$ different nanopores) for BSA and Kap95, respectively. The sharp decrease in event rate for BSA upon NupX coating of the pores indicates that BSA molecules are strongly hindered by the NupX meshwork formed inside the pore. In contrast, the transport rate of Kap95 through the coated pore is nearly unaffected when compared to the bare pore. From these experiments, we conclude that the user-defined NupX does impart a selective barrier, very similar to native FG-Nups[26,28,31], by allowing efficient transportation of Kap95 while hindering the passage of BSA.

**MD simulations of NupX-lined nanopores reveal their protein distribution and selectivity**. We used coarse-grained MD simulations (Methods) to understand the selective properties of NupX-lined nanopores as observed in our experiments. The 20 nm height of these nanopores is the same as the SiN$_x$ membrane thickness, while we vary the diameter from 15 to 70 nm. Multiple copies of NupX are tethered to the nanopore lumen by their C-terminal domain in an equilateral triangular lattice with a spacing of 5.5 nm, based on estimates obtained from the SPR experiments (Supplementary Fig. 3, Methods). We note that the geometrical confinement by the nanopore may affect the grafting distance on the concavely curved interior pore wall (parallel to the pore axis) as compared to the planar geometry[56]. Based on 6 μs of coarse-grained MD simulations, we obtained the protein density distribution in the $(r, z)$-plane (averaged over time and angle $\theta$) within a NupX-lined nanopore of 30 nm in diameter (Fig. 4b), similar in size as the translocation experiments.

High-density regions form close to the attachment sites (i.e., the four dots at each wall in Fig. 4b) and along the central axis of the nanopore. Since the triangulated lattice (comprising four rows) does not strictly exhibit a symmetry plane along the $z = 0$ axis, a slight asymmetry (<10% in terms of protein density) occurs between the top and bottom of the density map. From these data, we obtained a radial protein density profile, averaged over the pore height for the pore region ($|z| < 10$ nm, Fig. 4c), which exhibits a maximum of 230 mg/mL at the pore centre for the 30 nm NupX nanopore system and is insensitive to the aforementioned small asymmetry. This density agrees well with values in the range of 200–300 mg/mL observed in earlier computational studies of the yeast NPC central channel[34,42]. We attribute the central localization of the NupX proteins to the combination of repulsion between the high C/H ratio extended domains near the pore wall and attraction between the cohesive, low C/H ratio collapsed domains of opposing NupX proteins. Since the average density in the access region (10 nm < $|z|$ < 40 nm, Fig. 4c) is found to be low in comparison to the average density within the pore region, we conclude that the NupX proteins predominantly localize within the nanopore. When the grafting distance is perturbed by ~10% in either direction (Supplementary Fig. 17) to values of 5.0 or 6.0 nm, similar density profiles are obtained. So even though the experimental grafting distance might be somewhat larger for the nanopore compared to the planar brushes due to the different geometrical confinement, similar profiles would be expected for more sparsely coated pores.

The organization of the NupX proteins inside the nanopore geometry changes notably with pore diameter (Fig. 4d, e). For large diameter pores, the density profile of NupX proteins protruding from the pore surface quite well resembles that of a planar brush (cf. Supplementary Fig. 11), resulting in a central opening for pores that are ≥45 nm. When the pore diameter

reaches values <45 nm, NupX-coated nanopores are effectively sealed. This 45 nm length scale is remarkable, given the quickly decaying density profile of a planar NupX brush with a similar grafting distance (Supplementary Fig. 11). Upon further decreasing the pore diameter to values <25 nm, we find that the NupX collapsed domains are expelled from the pore region towards the access region, resulting in decreased densities in the central pore region (Fig. 4d, e). Interestingly, we find that these changes in NupX morphology as a function of pore diameter are in good qualitative agreement with predictions from earlier works on polymer-coated nanopores[57,58], which point to a curvature-dependent modulation of the brush height. More specifically, an increase in curvature (i.e., a decrease in pore diameter) of a concave brush substrate is expected to lead to a relative extension of the brush as compared to the planar geometry. In addition, attractive interactions between the cohesive head groups of NupX anchored at opposing pore walls will also contribute to the sealing of NupX pores. Finally, we note that a central opening in the NupX nanopore meshwork, present for diameters from 45 nm upwards, is consistent with the increased event frequency and translocation speed observed in large (60 nm) NupX-coated pores (Supplementary Fig. 15).

Using a relation between the local protein density and the local conductivity separately for the pore and access regions[31], we calculated the conductance of the NupX nanopores for varying diameters (Fig. 4d, Supplementary Fig. 18, Methods). The calculated conductance from the simulated NupX-lined pores is shown in Fig. 4f (black squares) together with the experimental conductances for bare and NupX-coated pores (open circles). Note that we adopted a critical protein density of 85 mg/mL from the earlier work on Nsp1[31] in our density–conductivity relation, but assume a different dependency of the local conductivity on the local protein density (Methods). Rather than assuming an abrupt complete blockage of conductance above the critical protein density of 85 mg/mL, we now use an exponential relation that provides a more gradual reduction in conductance with density. The necessity of a different density–conductance relationship indicates that the conductivity of the NupX nanopore meshwork depends non-linearly on the average protein density. Interestingly, the slope of the conductance–diameter curve for NupX-lined pores converges to that of bare pores already at relatively small pore sizes. This is due to the formation of a hole within the NupX meshwork (Fig. 4d) already in pores with diameters over 40 nm, rendering these effectively similar to bare nanopores of smaller diameter.

A spatial segregation of different types of FG-motifs, as was observed in recent computational studies[3,34], is not studied here explicitly. However, we find both types of FG-motifs localize similarly in the high-density central region within the NupX nanopore channel (Fig. 4g, h). From these distributions and the observed selective transport of these pores (Fig. 3e–g), we can infer that a spatial segregation of different FG-motifs is not required for selective transport.

Finally, in order to assess the selective properties of NupX-lined nanopores, we simulated a 30 nm diameter NupX-lined nanopore in the presence of ten Kap95 or ten inert particles. We released Kap95/inert particles in the access region at the top and recorded their location over 5 μs of simulation time (see Methods, Fig. 5c, d). The Kap95 particles entered and left the NupX meshwork and sampled the pore lumen by traversing in the $z$-direction (Fig. 5c). They localized preferentially at positions radially halfway between the central pore axis and the edge of the nanopore, where their time-averaged density distribution takes the shape of a concave cylindrical region, as is shown in Fig. 5a. Kap95 was found to be capable of (re-)entering and leaving the meshwork on either side (Fig. 5c). Since no external electric field

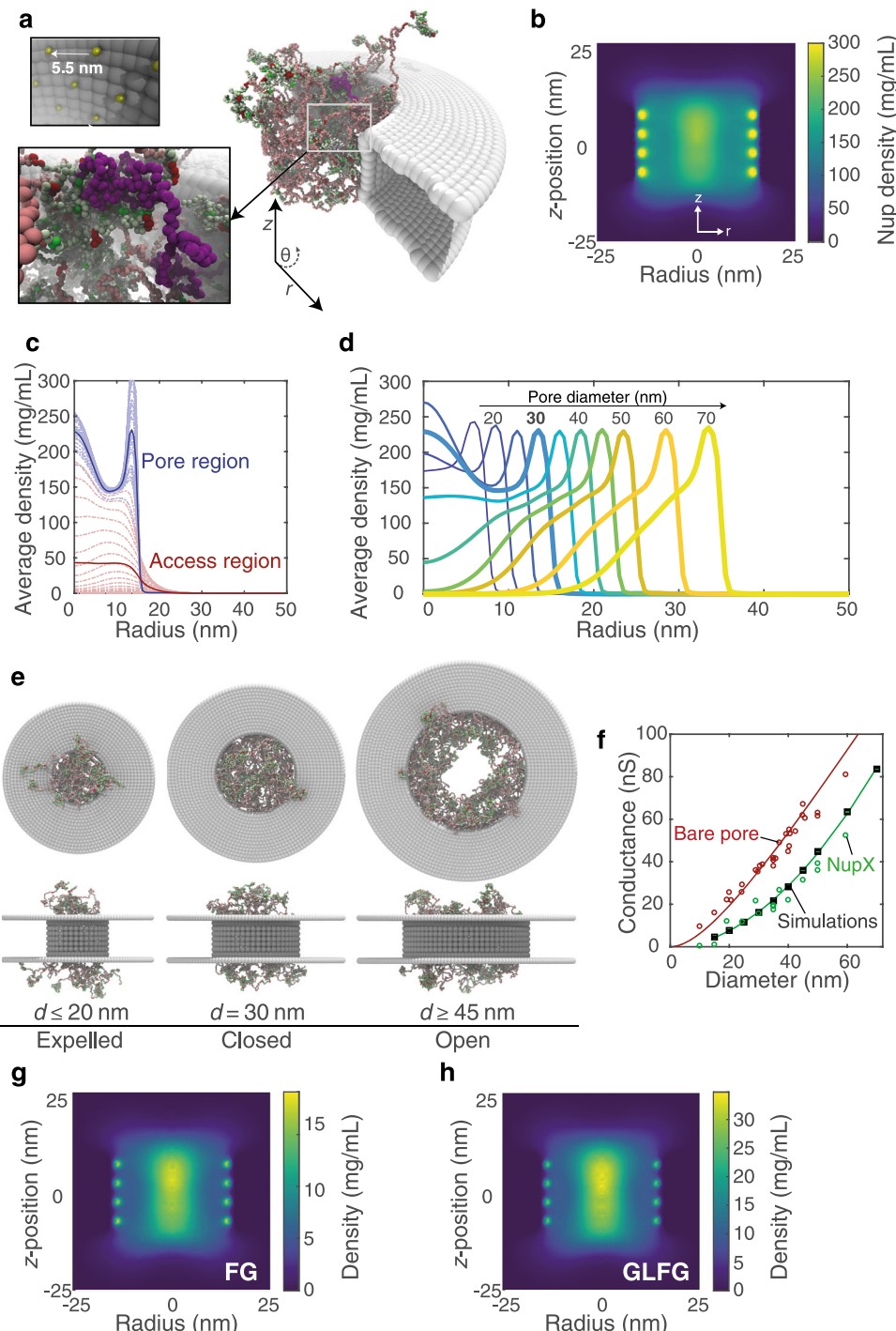

was applied, exiting and subsequent re-absorption of Kap95 into the NupX meshwork occurred and there was no directional preference for the motion of the Kap95 molecules, in contrast to the experiments. Interestingly, the NupX meshwork adapted itself to the presence of the Kap95 particles by expanding towards the access region (compare Fig. 4b and Supplementary Fig. 19): the protein density in the pore region decreased due to the presence of the Kap95, whereas the protein density increased in the access region. In contrast to the findings for Kap95, we observed that the inert particles, simulated under the same conditions, remained in the top compartment (Fig. 5b, d) and did not permeate into the NupX meshwork over the 5 μs time span of the simulation.

To quantify the selectivity of the 30 nm NupX-lined nanopores, we calculated PMF curves along the z-axis for both cargo types (Fig. 5e, see Methods). Kap95 experienced a negative free energy difference of approximately 8 kJ/mol, which amounts to a binding energy of just over $3\,k_BT$ per Kap95. On the other hand, inert particles experience a steep energy barrier of approximately 18 kJ/mol, which corresponds to over $7\,k_BT$ per protein. The obtained Kap95 free energy profiles are similar to those found in other simulation studies of cargo permeation through NPCs[8,47] or NPC-mimicking systems[31,59]. The Kap95 binding free energy differences along the nanopore axis are considerably smaller than the computed free energy profiles for NupX brushes (Fig. 2i). This

**Fig. 4 Protein distribution and conductance of NupX-coated pores. a** Snapshot of a biomimetic nanopore simulation. NupX proteins (following the colouring scheme of Fig. 1) were tethered with a grafting distance of 5.5 nm (yellow, top inset) to a cylindrical occlusion made of inert beads (grey). Pore diameters ranged from 15 to 70 nm, where the pore thickness was 20 nm throughout. Bottom inset: highlight of a single NupX protein (purple) within the NupX meshwork. **b** Axi-radial map (averaged over time and in the azimuthal direction) of the protein density within a 30-nm NupX-lined nanopore, from 6 μs simulations. Dark colours indicate regions of low density, brighter colours indicate regions of high density. The collapsed domains of the NupX proteins form a high-density central plug. The high-density regions near the pore radius (15 nm) coincide with the anchoring sites of the NupX proteins. **c** Density distributions (thick lines) for the pore (blue, $|z| < 10$ nm) and access (red, $10$ nm $< |z| < 40$ nm) regions. Dashed curves indicate the average density within 1-nm thick slices in the $z$-direction. **d** Radial density distributions ($z$-averaged) for NupX-lined nanopores with diameters ranging from 15 to 70 nm (darker and lighter colours denote smaller and larger diameters, resp.). The curve for 30 nm is emphasized. An increase in pore size beyond 30 nm leads to a decrease in the pore density along the pore's central channel. **e** Side-view and top-view visualizations of 20, 30, and 45 nm diameter NupX-lined nanopores. For nanopores with diameters smaller than 25 nm, the pore region density decreases due to an expulsion of the collapsed NupX domains towards the access region. For nanopore diameters larger than 40 nm, the pore density decreases and a hole forms. For nanopore diameters of 25–30 nm, the pore region is sealed by the NupX cohesive domains. **f** Conductance scaling for bare and NupX-coated nanopores. Open circles indicate conductance measurements for bare (red) and NupX-coated (green) pores. Squares indicate time-averaged conductance values obtained from MD simulations via a density–conductance relation (Methods). Error bars indicate the standard deviation in the conductance and are smaller than the marker. Second-order polynomial fits to the bare pore (experimental) and the simulated conductance values are included as a guide to the eye. **g, h** Axi-radial density maps for FG- and GLFG-motifs, respectively. Both types of motif localize in the dense central region, indicating that there is no spatial segregation of different types of FG-motifs in NupX-coated nanopores.

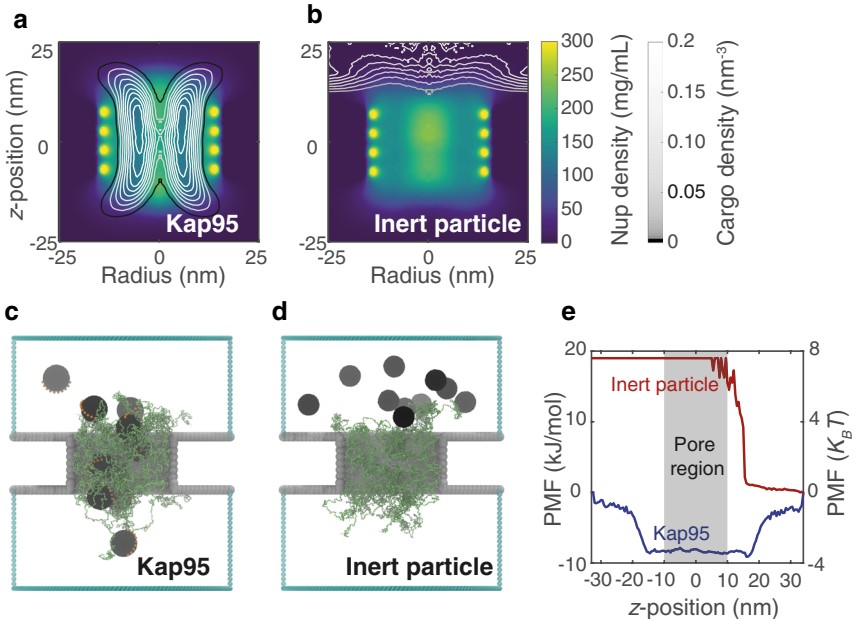

**Fig. 5 Effect of transporters on NupX-lined biomimetic pores. a** Contour graphs of the Kap95 number density (grey contours) superimposed on the NupX protein density distributions (in the presence of Kap95) within a 30 nm NupX-lined nanopore (NupX-density follows the same colouring scheme as in Fig. 4b and is shown separately in Supplementary Fig. 19). The protein meshwork adapts (as compared to the distribution in Fig. 4b) to accommodate the permeating Kap95 particles. **b** Density distribution of inert particles superimposed on the NupX protein density distribution in a 30 nm diameter NupX-lined nanopore. Inert particles remain in the top compartment and do not permeate the NupX protein meshwork. **c, d** Simulation snapshots of 30 nm NupX-lined nanopores in the presence of Kap95 particles (**c**, black spheres with orange binding spots) and inert particles (**d**, black spheres), which were released in the top compartment. Kap95 particles enter and exit the NupX meshworks at either side of the nanopore, whereas inert particles remain in the top compartment. **e** PMF curves of Kap95 (blue) and inert particles (red) along the $z$-coordinate, obtained via Boltzmann inversion of the normalized density profile along the $z$-axis. The pore region coincides with an energy well of over $3 k_B T$ for Kap95, whereas inert particles experience a steep energy barrier of $\sim 7 k_B T$.

most probably relates to the fact that the two studied reaction coordinates differ notably and cannot easily be compared: the reaction coordinate in Fig. 2i describes orthogonal entry into a brush that extends infinitely in the lateral direction, whereas the coordinate in Fig. 5e describes lateral entry and exit into the NupX assembly within the pore. As a result, one would not expect the free energy differences for transport through the nanopores to be similar to those obtained for entry into a brush geometry. Note that large free energy differences in our nanopores would also yield residence times that are orders of magnitude larger than the observed ~ms dwell times in our nanopore experiments[60]

(Fig. 3f). From our combined experimental and simulation results, we conclude that NupX-lined nanopores indeed reproduce the NPC's remarkable selectivity towards Kap95.

## Discussion

In this work, we introduced a 311-residue long artificial FG-Nup, termed NupX, that we rationally designed de novo based on the average properties of GLFG-type Nups (Nup49, Nup57, Nup100, Nup116, Nup145N) and which faithfully mimics the selective behaviour of the NPC. We experimentally found that substrates

coated with NupX brushes of varying grafting densities bind selectively to Kap95, while they did not interact with the control protein (BSA)s—a finding confirmed through coarse-grained MD simulations of the adsorption of Kap95 and inert particles. Consistent with these results, we found that Kap95 translocates through both uncoated and NupX-lined nanopores on a physiological (~ms) timescale[61], whereas BSA passage through the NupX-coated pores was effectively excluded. Coarse-grained MD simulations revealed how the NupX proteins form a dense (>150 mg/mL) phase that allows passage of Kap95 particles while excluding inert particles. Interestingly, we find that the high densities of the FG-rich NupX meshworks are comparable to those obtained in earlier simulation studies of yeast NPCs[42]. A comparison of the intrinsic protein density (i.e., the protein density of an individual molecule in solution, quantified by the mass per unit Stokes volume) of NupX (219 mg/mL) with that of Nsp1 (74 mg/mL) explains why our NupX meshworks have the tendency to localize more compactly inside nanopore channels than Nsp1 in earlier work[31]. The increased conductance of the denser NupX-lined nanopores (as compared to Nsp1) required a non-linear relation between the average protein density and the local conductivity, and indicates that the average protein density is not the only factor that describes conductivity; the dynamics of the unfolded proteins and the local charge distribution might be important as well.

The design strategy presented in this work allows us to assess the role of the AA sequence of the spacer regions in GLFG-Nups. Spacer residues were reported to be involved in the interaction interface of Nup-NTR complexes[62–64], highlighting a possible specific role of these domains in the binding of NTRs. In the current work, we assigned the positions of spacer residues along the NupX AA sequence entirely randomly, in both the collapsed, FG-rich low C/H ratio domain, and the extended high C/H ratio domain. This indicates that no specific spacer sequence motifs are required to facilitate the fast and selective transport of NTRs like Kap95. The consistency of the Stokes radii of different NupX designs within our simulations (Fig. 1h) supports this finding.

Furthermore, our results shed light on the functional role of the spatial segregation of FG- and GLFG-motifs that was observed in earlier work[3,34]. Although these recent computational studies observed such a feature and suggested that it plays a role in selective transport, the coinciding distributions of FG- and GLFG-motifs (Fig. 4g, h) show that no spatial segregation of different types of FG-motifs exists within our selective nanopores. Notably, this does not rule out a different functional role for the spatial segregation of different types of FG-motifs, which can be explored in future work.

The combined design and characterization approach presented here, with brush-adsorption and nanopore-transport measurements on the one hand and coarse-grained MD simulations on the other, provides a powerful and exciting platform for future studies of artificial FG-Nups: one can now start to systematically examine the relation between FG-Nup AA sequence and size selectivity of the NPC. Such studies could, e.g., entail the design of FG-Nups with radically different physicochemical properties (i.e., FG-spacing, FG-motif type, spacer domain C/H ratios, sequence complexity) to assess the selective properties of nanopore systems functionalized with these designer FG-Nups. Indeed, solid-state nanopores modified with a single type of FG-Nup were shown in this and other works[26,28,31] to reproduce NPC transport selectivity, justifying the use of a single type of artificial Nups within an environment structurally similar to the NPC. Moreover, in view of the similarity[5] and redundancy[21,35] of different FG-Nups within the NPC and the ability of our method to robustly reproduce FG-Nup properties (Fig. 1h and Supplementary Fig. 20), we are confident that a single artificial FG-Nup can

capture the selective barrier function of the NPC. However, given that even minimally viable native NPCs[21,35] contain several different FG-Nups, it is worth mentioning that NPC mimics with a heterogeneous set of (artificial) FG-Nups can be created as well: DNA origami scaffolds[33] potentially allow us to position different artificial FG-Nups with great control, thus enabling systematic studies of how the interplay of different (artificial) FG-Nups gives rise to various transport properties of the NPC.

Finally, the design procedure that we introduced here is not limited to applications in nucleocytoplasmic transport. It may, e.g., be possible to use a comparable approach to create de novo selective molecular filters (e.g., for use in artificial cells[65,66]), systems that would rely on selective partitioning of molecules in meshworks of unfolded proteins with assigned properties. Control can be asserted over the composition and geometry of the meshwork, e.g., by means of recently developed DNA origami scaffolds[32,33]. More generally, the approach illustrated here may enable future studies of the physical properties underlying phase separation of intrinsically disordered proteins[30]. One could, e.g., include degrees of freedom such as the proteins' second virial coefficient ($B_{22}$), or the charge patterning ($\kappa$), which have been linked to the phase behaviour of intrinsically disordered proteins[67,68]. We envision that just like the field of de novo protein design has come to fruition with improved understanding of protein folding[69], the design of unstructured proteins like NupX will enable a versatile platform to study the intriguing functionality of intrinsically disordered proteins.

## Methods

**Analysis of GLFG-Nups and design of synthetic Nups**. Protein sequences of *Saccharomyces cerevisiae* GLFG-type Nups (i.e., Nup100, Nup116, Nup49, Nup57 and Nup145N) were analyzed using a script custom-written with the R programming package (version 3.3.1). Following the definitions of high C/H-ratio and low C/H-ratio unfolded FG-Nup domains as given in Ref. [5], we obtained histograms of the AA frequencies in both the collapsed (low C/H-ratio) and extended (high C/H-ratio) domains. The collapsed/extended domain sequences of NupX were then assigned in three steps. First, the collapsed and extended domains of NupX were assigned equal lengths of 150 residues each. Then, by normalizing the distributions in Fig. 1d to the number of available residues within each domain, the total pool of AAs within each domain was obtained. Lastly, these AAs were randomly assigned a sequence index within each domain, with as a boundary condition the presence of FG and GLFG repeats spaced by ten residues within the low C/H-ratio domain. This approach was repeated iteratively in combination with disorder predictions using the online PONDR disorder prediction utility[36] until a sufficiently disordered design was obtained. The final version of the NupX AA sequence was also analyzed for secondary structure using DISOPRED[37,38] and Phyre2[39]. A 6-histidine tag was added to the N-terminus of the NupX sequence in order to facilitate protein purification (see Protein purification section). Finally, on the C-terminus a cysteine was included to allow the covalent coupling of the NupX protein to the surface.

**Expression and purification of NupX and Kap95**. The synthetic NupX gene (Genscript), appended with codons for an N-terminal His6-tag and a C-terminal cysteine residue, was cloned into pET28a and expressed in *Escherichia coli* ER2566 cells (New England Biolabs, *fhuA2 lacZ::T7 gene1 [lon] ompT gal sulA11* R(mcr73:: miniTn10–TetS)2 *[dcm]* R(zgb-210::Tn10–TetS) *endA1* Δ(mcrCmrr)114::IS10). To minimize proteolysis of NupX, the cells were co-transformed with plasmid pED4, a pGEX-derivative encoding GST-3C-Kap95 under control of the tac promoter. Kap95 was expressed as a C-terminal GST fusion protein in *Escherichia coli* ER2566 cells from plasmid pED4, a pGEX-derived construct in which the thrombin cleavage site was replaced by a 3C protease cleavage site using primers ed7 and ed8 (Supplementary Table 4). Cells were cultured in shake flasks at 37 °C in Terrific Broth supplemented with 100 μg/mL ampicillin and 50 μg/mL kanamycin, and expression was induced at OD600~0.6 with 1 mM IPTG. After 3 h of expression the cells were harvested by centrifugation, washed with PBS, resuspended in buffer A1 (50 mM Tris/HCl pH 7.5, 300 mM NaCl, 8 M urea, 5 mg/mL 6-aminohexanoic acid supplemented with one tablet per 50 mL of EDTA-free cOmplete ULTRA protease inhibitor cocktail) and frozen as 'nuggets' in liquid nitrogen. Cells were lysed with a SPEX cryogenic grinder, after thawing 1,6-hexanediol was added to a final percentage of 5%, and the lysate was centrifuged for 30 min at 1250,00 × g in a Ti45 rotor (Beckman Coulter). The supernatant was loaded onto a 5 mL Talon column mounted in an Akta Pure system, the column was washed with buffer A2 (50 mM Tris/HCl pH 7.5, 300 mM NaCl, 800 mM urea, 5 mg/mL 6-aminohexanoic acid, 2.5% 1,6-hexanediol) and NupX was eluted with a

linear gradient of 0–200 mM imidazole. Peak fractions were pooled, diluted tenfold with buffer A2 lacking sodium chloride, loaded onto a 1 mL HiTrap SP sepharose HP column and NupX was eluted with a linear gradient of 0–1 M NaCl.

Kap95 was expressed as a C-terminal GST fusion protein in *Escherichia coli* ER2566 cells from plasmid pED4, a pGEX-derived construct (kindly provided by Jaclyn Novatt) in which the thrombin cleavage site was replaced by a 3C protease cleavage site. Cells were grown in shake flasks at 30 °C on LB medium supplemented with 100 μg/mL ampicillin, induction was induced at OD600~0.6 with 1 mM IPTG, and growth was continued overnight. Cells were harvested by centrifugation, washed with PBS, resuspended in TBT buffer (20 mM HEPES/NaOH pH 7.5, 110 mM KOAc, 2 mM MgCl2, 0.1% (w/v) Tween20, 10 μM CaCl2 and 1 mM β-mercaptoethanol), and lysed by a cell disruptor (Constant Systems) at 20 kpsi. Following centrifugation for 30 min at 125,000 × g in a Ti45 rotor (Beckman Coulter), the supernatant was loaded onto a 2 mL GSTrap 4B column mounted in an Akta Pure system. The column was washed with TBT buffer, TBT + 1 M NaCl and TBT + 0.1 mM ATP, and the fusion protein was eluted with TBT + 10 mM reduced glutathione. The GST moiety was cleaved off by overnight digestion with home-made 3C protease, and Kap95 was separated from GST and the protease by size exclusion chromatography on a Superdex S200 column pre-equilibrated with TBT buffer.

**QCM-D sample preparation and data acquisition.** QSense Analyzer gold- and SiN-coated quartz QCM-D chips were purchased from Biolin Scientific, Västra Frölunda, Sweden. Prior to the experiment, chips were immersed in RCA-1 solution, which consisted of 30% Ammonium Hydroxide, 30% Hydrogen Peroxide and deionized (DI) water in 1:1:5 ratio, for ~30 min at 75 °C. This step was used to clean the surface from carbon species, as well as to enrich the surface with hydroxyl groups in case of the SiN-coated chips. Chips were further rinsed with DI water, sonicated for ~10 min in pure ethanol and blow-dried with a nitrogen stream. Before each experiment, flow-cells were disassembled, cleaned by sonication for 20–30 min in freshly prepared 2% SDS, rinsed with DI water and blow-dried with a nitrogen stream. For SiN-coated quartz sensors, the SiN surface was chemically engineered in order to add free maleimide groups (see Preparation of NupX-coated nanopores for details). Prior to the coating of the gold surface, protein solutions containing either NupX, Nsp1 or Nsp1-S were incubated in PBS with 1mM TCEP for at least half an hour, which was also present during the coating step. Similarly, MUTEG solutions also included 10 mM TCEP.

QCM-D data were monitored and recorded with sub-second resolution using Qsoft, which was provided by the company together with Qsense Analyzer. Buffer was injected into the flow-cell chamber at a constant flow-rate of 20 μL/min using a syringe pump. Experiments were all performed at room temperature. Shift in the resonance frequency ($\Delta f$) and dissipation ($\Delta D$) can be, in first approximation, attributed to changes in deposited mass and viscoelastic properties of the film, respectively. $\Delta f$ and $\Delta D$ were acquired at the fundamental tone ($n = 1$) and the five overtones ($n = 3, 5, 7, 9, 11$). The normalized second overtone $\frac{\Delta f_5}{5}$ was used for display and analysis. Data processing and plotting were performed using a custom-written Matlab script.

**SPR measurements and analysis.** All measurements were performed using a Bionavis MP-SPR Navi™ 220A instrument equipped with two 670-nm laser diodes focused on two different spots on the sample surface. Both gold- and silicon dioxide-coated sensor slides were used (Bionavis). Grafting of proteins or MUTEG to the gold- and silica-coated sensors was performed using the same protocol as for the gold-coated QCM-D chips and for the nanopore chips, respectively. After the final incubation step, the chips were rinsed with milliQ water, ethanol and gently blow-dried with pure nitrogen. Immediately before measurements of the sensors in air, the backside of each sensor was cleaned by carefully rubbing lint-free lens tissue soaked in 2-propanol (Sigma Aldrich) followed by blow-drying both sides with nitrogen. At least two repeat measurements were performed for all sensor slides to verify no signal drifting occurred due to adsorption of moisture. The background signal from each sensor was measured before grafting the adlayers to the sensor surfaces and was used as a starting point for modelling the adlayer thickness, $d$. The silicon dioxide thin film thickness was evaluated to $15.1 \pm 0.5$ nm across the different samples and surface spots prior to surface grafting. Three replicate sensor slides were used for each type of adlayer (except for MUTEG which used one sensor slide per concentration measurement).

Adlayer thickness was determined from least-square fitting measurements with Fresnel models using an approach similar to those used in previous works[70,71], which we briefly describe here. The following information was used to obtain adlayer thickness: a fit performed on the angular reflectivity spectra close to the resonance angle (see Supplementary Fig. 3), knowledge of the sensor layer thicknesses and refractive indices[72], the bulk refractive index of air in the measurement cell, as determined from the total internal reflection angle of each measurement and the assumption that the adlayer refractive index $n$ is the same as its dry bulk counterpart ($n_{protein} = 1.53$[73], $n_{APTES} = 1.42$, $n_{MUTEG} = 1.456$[74]). The formed adlayers were further assumed to be homogeneous and not containing solvent and to be free of contaminants. Grafting densities, $\Gamma$, of MUTEG, Nsp1, Nsp1-S and NupX were obtained from the Fresnel model determined $d$ using

the relation:

$$\Gamma = \frac{\rho d N_A}{M} \quad (1)$$

where $\rho$ is the density ($\rho_{MUTEG} = 1.09$ g/cm, $\rho_{protein} = 1.35$ g/cm³; Ref. [70]), $N_A$ is Avogadro's constant, and $M$ is the molecular weight ($M_{MUTEG} = 380$ Da, $M_{Nsp1} = 65.7$ kDa, $M_{Nsp1-S} = 62.1$ kDa, $M_{NupX} = 32.5$ kDa).

**Preparation of NupX-coated nanopores and current data acquisition.** Solid-state nanopores with diameters from 10 to 60 nm were drilled using TEM in glass-supported SiN$_x$ free-standing membranes. Glass chips were purchased from Goeppert. We refer to Ref. [46,58] for details on the fabrication of the chip substrate and free-standing membrane. Freshly drilled solid-state nanopores were rinsed with ultrapure water, ethanol, acetone, and isopropanol, followed by 2–5 min of oxygen plasma treatment, which was performed in order to further clean and activate the nanopore surface with hydroxyl groups. Next, chips were incubated in 2% APTES (3-aminopropyl-triethoxysilane) (Sigma Aldrich) in anhydrous toluene (Alfa Aesar) for 45–60 min at room temperature, shaking at 400 rpm, followed by 15 min in anhydrous toluene for washing. These two steps were performed in a glove-box under constant nitrogen stream in order to prevent the APTES from polymerizing. Then, chips were further rinsed with ultrapure water, ethanol, and heated at 110 °C for at least 30 min. This step was used to fixate the APTES layer by favouring further binding between the unreacted ethoxy groups.

The nanopore surface was thus coated with primary amines, which were subsequently reacted to Sulfo-SMCC (sulphosuccinimidyl-4-(N-maleimidomethyl)-cyclohexane-1-carboxylate) (2 mg no-weight capsules (Pierce)), a crosslinker that contains NHS-ester (reacts to amines) and maleimide (reacts to thiols) groups at opposite ends, for >3 h at room temperature, shaking at 400 rpm. Chips were subsequently washed in PBS for 15 min and incubated with thiolated proteins for 2–3 h, which were pretreated with 5 mM TCEP for ~30 min in order to reduce the thiol groups. Chips were further washed in PBS before the electrical measurement. Raw ionic current traces were recorded at 100 kHz bandwidth with an Axopatch 200B (Molecular devices) amplifier, and digitized (Digidata 1322A DAQ) at 250 kHz. Traces were monitored in real time using Clampex software (Molecular devices). Data were digitally filtered at 5 kHz using a Gaussian low-pass filter and analyzed using a custom-written Matlab script[75].

**Dynamic light scattering (DLS) measurement of the hydrodynamic diameter.** DLS experiments were performed using Zetasizer Nano ZS (Malvern). Cuvettes of 100 μL (Brand GMBH) were used for the measurement. All protein hydrodynamic diameters were measured in 150 mM KCl, 10 mM Tris, 1 mM EDTA, at pH 7.5, and averaged over three experiments. Mean value and standard deviation for each of the proteins used are reported in Supplementary Table 1. Proteins that contained exposed cysteines (NupX, Nsp1 and Nsp1-S) were pretreated with TCEP (present in at least 100× excess) in order to break disulfide bonds.

**Coarse-grained model for unfolded proteins.** All coarse-grained MD simulations were performed using our earlier developed 1BPA model for unfolded proteins[42,76]. This model maps complete AAs to single beads with a mass of 124 amu placed on the $C_\alpha$ position, separated by an average bond length (modelled as a stiff harmonic potential) with an equilibrium distance of 0.38 nm. Backbone potentials were assigned via an explicit coarse-grained mapping of Ramachandran data of a library of the coil regions of proteins that distinguishes flexible (i.e., Glycine), stiff (i.e., Proline) and regular AAs[76]. Non-bonded interactions between different AA residues are based on their respective hydrophobicity (normalized between 0 and 1 and based on the free energy of transfer between polar and apolar solvents) and obey the following interaction potential[42]:

$$\Phi_{HP} = \begin{cases} \epsilon_{rep}\left(\frac{\sigma}{r}\right)^8 - \epsilon_{ij}\left[\frac{4}{3}\left(\frac{\sigma}{r}\right)^6 - \frac{1}{3}\right], & r \leq \sigma \\ (\epsilon_{rep} - \epsilon_{ij})\left(\frac{\sigma}{r}\right)^8, & \sigma \leq r \end{cases} \quad (2)$$

where $\epsilon_{rep} = 10$ kJ/mol and $\epsilon_{ij} = 13 \cdot \sqrt{(\epsilon_i \epsilon_j)^\alpha}$ kJ/mol, with $\epsilon_i$ the normalized (between 0 and 1) hydrophobicity of a residue $i$ and $\alpha = 0.27$ a scaling exponent. The electrostatic interactions within the 1BPA model are described by a modified Coulomb law:

$$\Phi_{EL} = \frac{q_i q_j}{4\pi\epsilon_0 \epsilon_r(r) r} \exp(-\kappa r), \quad (3)$$

where the electrostatic interactions are modulated via a Debye screening component. This form of electrostatics takes into account the salt concentration (set at 150 mM here, via a screening length $\kappa = 1.27$ nm$^{-1}$) together with a solvent polarity at short distances via a distance-dependent dielectric constant:

$$\epsilon_r(r) = 80\left[1 - \frac{r^2}{z^2}\frac{e^{\frac{r}{z}}}{\left(e^{\frac{r}{z}} - 1\right)^2}\right], \quad (4)$$

where $z = 0.25$. Non-bonded interactions are cut-off at 2.5 nm (hydrophobic interactions) or 5.0 nm (electrostatic interactions). Since the 1BPA model operates

without explicit solvent, we apply stochastic dynamics with a coupling time $\tau_T$ of 50 ps. Stochastic dynamics handles temperature coupling implicitly, ensuring that the system operates within a canonical ensemble at a reference temperature of 300 K. We refer the reader to the original work for further details on the used 1BPA model[42,76]. Unless otherwise mentioned, all simulations were performed using the above forcefield and corresponding settings, employing the GROMACS[77] molecular dynamics software (version 2016.1/2016.3) on a parallelized computer cluster. A complete overview of all simulations in this work is provided in Supplementary Table 3.

**Calculating Stokes radius of NupX and variations.** Intrinsically disordered proteins were modelled using the 1BPA model[42,76], starting from an extended configuration. After energy minimization (steepest descent) and a brief (5 ns) equilibration step, we simulated the individual proteins for $5 \times 10^8$ steps using a timestep of 20 fs (total simulation time: 10 μs). Conformations were extracted every 10,000 frames (i.e., every 200 ps). In order to calculate the Stokes radii ($R_s$) from the MD trajectories, we extracted protein conformations every 2 ns and applied the HYDRO++ software[78] in order to calculate the $R_s$ values. This procedure yields a total of 5000 Stokes radii per protein.

**Calculating NupX brush density profiles and PMF curves of cargo adsorption.** We modelled the brush substrate as a fully triangulated (sometimes denoted as hexagonally) closed-packed array of sterically inert beads with a diameter of 3 nm. NupX proteins were tethered on top of the scaffold by their C-terminus, following an equilateral triangular lattice with a uniform grafting distance of 4.0 or 5.7 nm. A fully triangulated lattice is close-packed in two dimensions, meaning that a unique length scale sets the grafting density. The simulation box consisted of a $24 \times 24 \times 81.5$ nm³ triclinic and fully periodic unit cell (Fig. 2h). In our simulation of a less dense brush (grafting distance of 5.7 nm), the simulation box size was accordingly scaled up to 34.2 nm in the lateral dimensions. The grafting pattern of the NupX proteins was placed such as to ensure homogeneity of the NupX brush in the lateral plane throughout the periodic boundaries. Density profiles for the NupX brushes and FG/GLFG repeats were obtained by simulating the NupX brush systems for $1.75 \times 10^8$ steps (3.5 μs) using a timestep of 20 fs. The first 500 ns of the simulation trajectory was discarded as equilibration. We modelled Kap95 and inert particles in the following way: the Kap95 particle consists of sterically repulsive beads, arranged in a geodesic shell such that the particle has a diameter of 8.5 nm, consistent with the hydrodynamic dimensions of the Kap95 protein (Supplementary Table 1). The Kap95 surface beads interact with the NupX AA beads through the repulsive term of $\Phi_{HP}$ (i.e., volume exclusion), and the modified Coulomb potential $\Phi_{EL}$, where we distributed the charge (total net charge of −43e) of Kap95 over the Kap95 surface beads. This modelling choice is based on the high degree to which charged residues are exposed on the surface of Kap95[63]. We preserved the structure of the particle by applying a harmonic restraint of 40,000 kJ/mol on bead pairs whenever the distance between beads within the reference structure was below a cut-off of 1 nm. A total of ten hydrophobic binding pockets were placed at a mutual distance of 1.3 nm along an arc (Supplementary Fig. 12) on the surface of the Kap95 particle[8,31,46,47]. The binding sites interact with NupX AA beads via the hydrophobic potential $\Phi_{HP}$, where the hydrophobicity of these binding sites was set equal to that of Phenylalanine. In the same way as for the Kap95, we assembled an inert particle[10,31,47] with a diameter similar to inert control proteins (7.5 nm, Supplementary Table 1). Other than steric repulsion, no specific interactions between the inert particle on the one hand, and the AA or substrate beads on the other were assigned. Using a harmonic restraint of 100 kJ/mol, we generated umbrella sampling windows by pulling the cargo in the negative $z$-direction (while freezing the particle's movement in the $xy$-plane) with a pulling velocity of −0.001 nm/ps and a timestep of 20 fs along the centre of the triclinic box. Starting configurations were extracted every 0.5 nm, yielding 51 umbrella windows per cargo. After energy minimization (removal of overlap between beads) via the steepest descent algorithm, we performed 100 ns ($5 \times 10^6$ steps) of equilibration, and 1 μs ($5 \times 10^7$ steps) of production MD per umbrella window, where the Kap particles were restrained using a harmonic umbrella potential of 100 kJ/mol in the $z$-direction, applied to the cargo's centre of mass. Aside from this restraint, the particles were free to rotate and move in the $xy$-plane. PMF curves were obtained using the WHAM via the g_wham[48] utility of GROMACS.

**Coarse-grained MD simulations of NupX-lined nanopores.** We modelled the SiN nanopores as cylindrically shaped occlusions in a membrane constituted entirely of sterically repulsive beads with a diameter of 3 nm. The height of the nanopore was 20 nm in all cases, with diameters ranging from 15 to 70 nm. NupX proteins were modelled using the 1BPA model described earlier and tethered to the inner surface of the cylinder in an equilateral triangular lattice with a grafting distance of 5.5 nm. This value was estimated from NupX immobilization on silica surfaces using the SPR technique and should be taken as a lower limit given that molecular adsorption in nanopores can be influenced by steric repulsion and geometric confinement[56]. Simulations were carried out for $2 \times 10^8$ steps using a timestep of 15 fs (3 μs), or $4 \times 10^8$ steps (6 μs) for the single case of a 30 nm diameter NupX-lined nanopore.

**Density distributions and nanopore conductance from nanopore simulations.** Axi-radial density maps were obtained from NupX nanopore simulation trajectories using the 'gmx_densmap' utility of GROMACS, where a bin size of 0.5 nm was used to construct number densities within a cylinder centred on the nanopore. Average densities were extracted for the pore and access regions by averaging the axi-radial density distributions over the coordinate ranges $|z| \leq 10$ nm and 10 nm $< |z| < 40$ nm, respectively[31].

The conductance of NupX-lined nanopores was obtained by assuming that the conductance $G(d)$ is governed by a modified Hall-formula[31,53]:

$$G(d) = \left( \frac{4l}{\sigma_{pore}\pi d^2} + \frac{1}{\sigma_{access}d} \right)^{-1}, \tag{5}$$

where $l = 20$ nm is the height of the nanopore, $d$ denotes the diameter (15–70 nm), and $\sigma_{pore}$ and $\sigma_{access}$ denote the conductivities in the pore and access regions, respectively. The conductivities in both regions can be extracted from the axi-radial density distributions by integrating and normalizing the local conductivity over the pore diameter and corresponding height ranges:

$$\sigma_{pore} = \frac{1}{\pi l \frac{1}{4}d^2} \int_{z=-\frac{l}{2}}^{z=+\frac{l}{2}} \int_{r=0}^{r=\frac{d}{2}} 2\pi \sigma_{bulk} r\sigma(r,z) dr dz \tag{6.1}$$

$$\sigma_{access} = \frac{1}{\pi h \frac{1}{4}d^2} \int_{z=\frac{l}{2}}^{z=\frac{l}{2}+h} \int_{r=0}^{r=\frac{d}{2}} 2\pi \sigma_{bulk} r\sigma(r,z) dr dz \tag{6.2}$$

The local conductivity $\sigma(r,z)$ follows from the local axi-radial density distribution $\rho(r,z)$:

$$\sigma(r,z) = \exp\left( -\sqrt[3]{\frac{\rho(r,z)}{\rho_c}} \right), \tag{7}$$

where $\rho_c$ is set to 85 mg/mL[31]. Whereas this relation is a zero-parameter fit, the dependency of the conductivity on the local protein density is different from the linear model with a strict cut-off used in earlier work[31]. Here, the conductivity drops quickly with protein density initially while decreasing only slowly at high protein densities. This change in dependence was necessary since NupX-lined pores show a higher conductance at higher densities than the NPC mimics in earlier work. Axi-radial density distributions and the corresponding conductance were calculated for 100 ns windows to obtain an average conductance for each pore size. The sensitivity of this method was tested against the time averaging window of the density distributions and was found to only be marginally influenced by the window size, nor did the conductance change with time.

**Selectivity of NupX-lined nanopores.** We probed the selectivity of a NupX-lined nanopore with a diameter of 30 nm by inserting either ten Kap95 molecules or ten inert particles to the *cis*-side of the nanopore. To speed up sampling, Kap95 or inert particles were confined to the periphery of the nanopore using a cylindrical constraint that prevents Kap95 or inert particles from entering regions with $|z| > 40$ nm or $r > 35$ nm[79]. This occlusion consisted entirely of sterically repulsive beads with a diameter of 3 nm, which only interact with Kap95 or inert particles. The size of the cylindrical constraint was set such that Kap95 molecules or inert particles can only leave the access region by entering the NupX-lined nanopore. The total simulation time spanned $3.33 \times 10^8$ steps (5 μs), where we discarded the first 10% (500 ns) of the trajectory as equilibration. Axi-radial density maps for the protein density and contour graphs for the Kap95 and inert particle densities were obtained using the 'gmx_densmap' utility using a bin size of 0.5 nm. We reported a cargo number density in lieu of a mass density, since the number of beads that constitute the Kap95 or inert particles does not necessarily correspond to the number of AAs in either protein.

To calculate the PMF curve along the $z$-axis for Kap95 and inert particles, we integrated the axi-radial number density of the centre of mass of the particles in the radial direction, resulting in a one-dimensional axial number density $n(z)$. Normalizing $n(z)$ against the number of bins in the $z$-direction results in a probability distribution $p(z)$, from which the PMF can be calculated by using the inverse Boltzmann relation:

$$\text{PMF}(z) = -k_B T \ln p(z). \tag{8}$$

The PMF curves for both Kap95 and inert particles (Fig. 5e) were shifted such that the curves were zero at $z = 35$ nm. Regions with zero density (leading to divergence of the $\ln p(z)$-term) were set equal to the maximum of the PMF curve.

**Reporting summary.** Further information on research design is available in the Nature Research Reporting Summary linked to this article.

## Data availability
Data supporting the findings of this manuscript are available from the corresponding authors upon reasonable request. A reporting summary for this article is available as a Supplementary Information file. Source data are provided with this paper.

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

## Acknowledgements

We would like to thank the Görlich Lab for sharing purified Nsp1 and Nsp1-S proteins, and Jacklyn Novatt for the protocols on FG-Nup purification that we used for our artificial FG-Nup. We thank Meng-yue Wu for technical assistance on the TEM. This research was funded by NWO-I programme 'Projectruimte', grant no. 16PR3242-1. We acknowledge useful discussions with Paola de Magistris, Adithya Ananth, Wayne Yang, Sergii Pud, Daniel Verschueren, Sonja Schmid and thank Ralf Richter for providing useful comments on the manuscript. H.W.d.V. acknowledges support from the CIT of the University of Groningen and the Berendsen Centre for Multiscale Modeling for providing access to the Peregrine and Nieuwpoort high performance computing clusters. C.D. acknowledges support from the ERC Advanced Grant SynDiv (no. 669598) and the NanoFront and BaSyC programmes.

## Author contributions

A.F. and C.D. devised the experiments. H.W.d.V., E.v.d.G. and P.R.O. devised the simulations. E.O.v.d.S. cloned and purified the proteins. A.F. carried out the QCM-D and nanopore experiments and analysis. J.A. and A.D. carried out the SPR experiments and analysis. H.W.d.V. carried out the simulations and analysis. A.F., H.W.d.V., P.R.O. and C.D. wrote the manuscript.

## Competing interests

The authors declare no competing interests.
