## [Peer Review File · Nature Communications]

Reviewer #1 (Remarks to the Author):

This work by Fragasso et al. tests whether it is possible to rationally design a protein capable of recapitulating the selective sorting properties of the “FG” nups. The selective transport mechanism of the nuclear pore complex (NPC) remains a pressing and fundamental problem in cell biology. Through a clear articulation of design parameters and a compelling combination of MD simulations and experimental data, the authors make a strong case for the successful generation of “NupX”. As shown, NupX is capable of binding to the karyopherin Kap95 with an affinity similar to other FG-nups and, most strikingly, exhibits clear selective translocation through NupX-coated nanopores. These data, like all of the data presented in the manuscript, are of excellent quality. Thus, this paper is essentially a proof-of-concept study that opens up new doors to investigate the fundamental properties of the FG-nups that ultimately underlie macromolecular sorting in ways that could be argued are much more controlled than within *in vivo* settings. It is rather unfortunate, however, that the authors did not attempt to answer any biology-inspired question as the weakness of the work is that it provides little new information about the selective sorting mechanism itself. The one exception here may be the role of so-called spacer regions between the FG repeats that have been suggested by some to impact selectivity; the data presented here suggest that they are not important. A suggestion may be to place this finding in a more prominent position in the paper (e.g. the abstract) to broaden the impact of this work to more biologically minded readers.

Reviewer #2 (Remarks to the Author):

In this work, the authors present a strategy for studying nuclear transport, and specifically address the question of whether it is possible to build a synthetic protein that mimics the selective behavior of FG-nups (nucleoporins with FG repeats). They first design a synthetic nucleoporin, NupX, using rational design rules to preserve functionality. As an initial characterization, NupX is compared with other Nups by their amino acid position and content, charge/hydrophobic AA ratios, and disorder scores. Next, they use QCM-D to measure binding affinities of Kap95 (a nuclear transport receptor) to NupX as well as Nsp1 (a native nucleoporin) and Nsp1-s (a mutated nucleoporin) as positive and negative controls, respectively. Additionally, they use molecular dynamics simulations to predict the Kap95/tCherry absorption behavior. After determining that NupX behaves similar to native nucleoporins, they perform single-molecule translocation experiments with NupX-coated nanopores. This aspect of the work is a continuation of “Single-molecule transport across an individual biomimetic nuclear” (Nature Nanotechnology, 2011 Kowalczyk, et al.). BSA is shown to be mainly blocked by the coated pore whereas Kap95 passes through at high event rate. The authors subsequently use coarse-grained molecular dynamics simulations to predict the protein distribution and conductance of a NupX-lined nanopore of different diameters. Lastly, they simulated a NupX-lined nanopore in the presence of both Kap95 and tCherry (control) particles, showing that tCherry is largely excluded whereas Kap95 enters the pore.

The work is very well written, and the results are well-supported by experiment and simulations. In comparison to the former work (referenced above), this one presents sophisticated molecular dynamics simulations, as well as a novel application (i.e. the “first demonstration of a fully artificial nucleoporin”). Design of synthetic nucleoporins and testing in this manner can be used to elucidate the relationship between the FG-nup amino acid sequence and size-selectivity of the NPC. I have listed below a number of key points that could enhance the manuscript, and some suggestions to

improve the robustness of the conclusions. I would encourage the authors to consider them in a revision before a recommendation for publication in Nature Communication is made:

- The authors have previously shown tethering of FG-Nups to solid-state nanopores (Kowalczyk et al., Nature Nanotechnology 2011), and low-resolution MD simulations for such systems are routinely published. Therefore, the novelty of this work lies mainly in the development of artificial Nups. However, it is not clear why synthetic Nups such as NupX are needed to study the size selectivity of natural Nups. I would expect this development to have led to additional insights into the interactions between different nucleoporins, or to have highlighted regions of artificial Nups that have surprising or poorly understood functions. It would be useful to comment on this or emphasize the significance of the current study.
- Using solid-state nanopores is a nice way to simulate a pore complex in a controlled way, but to what extent is this an accurate model system for natural Nups in an NPC? The conclusion states that “(...) our artificial Nups can be used to build in vitro replicas of the NPC that are suitable for disentangling the NPC transport mechanism”, but this is by no means obvious. I would imagine the interaction between different Nups determines their collective transport behavior; it is not clear how the results shown here can be extended to simulate that. How would one go about functionalizing a solid-state pore with several different synthetic Nups in a way that accurately represents NPCs?
- The MD simulations show selectivity for Kap95, and translocations occur without an applied electric field. How does this compare with translocations through the nuclear membrane, and translocations under an external electric field in a solid-state pore? Will the orientation and organization of NupX be affected by an external electric field?
- Despite their relative simplicity (single-aa resolution), the MD simulations qualitatively predict the selective transport of Kap95 through a NupX-coated pore. However, they do not provide much insight beyond the experimental results, and therefore the utility of such simulations is questionable for more complex NPCs. I wonder if there is more to be learned from the extensive MD simulations ?
- The MD simulations show that the size selectivity is strongly dependent on the diameter of the pore. In this context, an experimental confirmation for one other pore size (at least) is missing.
- It seems like all the experimental results of translocations are taken from a single nanopore. Given the reliance on these results and, no doubt, the pore-to-pore variation in coating with NupX, I would encourage the authors to include some more statistics on these translocations.

Minor comments:

- The characterization of binding kinetics (Fig. 2) can be moved to the SI.
- Define NTR in introduction - nuclear transport receptor
- Define QCM-D in introduction- quartz crystal microbalance with dissipation monitoring
- “Whereas PONDR predicts one short folded segment between residues 189 and 209” - perhaps state also the normalized AA position (~ 0.65) as its presented in Figure 1g.
- “assuming an equilateral triangle lattice” – where does this assumption come from, which is also used in the molecular simulation?
- Figure S2- “estimated of” to “estimated as”
- “Importantly, the data thus show that the” – should be “shows”

- “z-values ranging from 2 tot 10 nm” –typo
- Figure 4f- “Open circles indicate measured the conductance for...” should be “open circles indicate conductance measurements for...”

Reviewer #3 (Remarks to the Author):

The paper by Fragasso et al nicely demonstrates the capacity of using sequence-designed artificial proteins to build biomimetic nanopore with high molecular selectivity. The substantial experimental and simulation results show that the designed NupX supports fast translocation of transport receptor Kap95 while blocking inert BSA. The experimental system has the potential of serving as a platform to study different FG-Nups of the NPC and to test the performance of sequence-controlled artificial polymers (J. Am. Chem. Soc. 2017, 139, 18, 6422-6430). We think the paper deserves to be published. Below are our questions:

1. The QCM experiment seems to suggest that the Nup films are very rigid. Are there any explanations for this observation?
2. What are the novel scientific insights from this work compared to the previous work with similar system setup from the same authors (eLife 2018;7:e31510)?
3. Our recent paper as cited by the authors (bioRxiv 568865 (2019). doi:10.1101/568865, now in press by the biophysical journal) shows that different FG groups occupy different spatial territories, suggesting that they have distinct/complementary functions. In the NupX designed by the authors, there are both GLFG and FG repeats. Have the authors investigated the different effects of these two FG groups on the structure and function of the FG-Nups?
4. How did the authors choose the interaction parameter between NupX and Kap95 in their simulations?

For transparency, this review is written by Igal Szleifer and Kai Huang from Northwestern University.

Reviewer #4 (Remarks to the Author):

The authors set out to fabricate synthetic pores that reproduce the selectivity of macromolecular binding and permeation in the nuclear pore complex (NPC). This is a challenging goal which, if successfully accomplished, should be of interest to a broad readership in the nanosciences, bottom-up synthetic biology, and possibly other research areas.

The manuscript is based on a novel method for the design of artificial FG nups. The rationale behind the method, described in the first part of the Results section, appears sensible overall. Whilst the exact method is a matter of taste to some extent in light of the various physical models of selective permeability that have been proposed, it is a good starting point for bottom-up reconstitution efforts.

The authors' approach to combine data from two distinct geometries (i.e. FG nups grafted either to a planar surface, or to the inner wall of a nanopore) and to correlate experimental data with coarse-grained molecular dynamics (MD) simulations for each of these two geometries, has its merit: The nanoscale analysis of assemblies of intrinsically disordered proteins remains technically difficult, and

the combination of experiment and theory is a promising approach to tackle this challenge as already shown by previous work in the field.

Unfortunately, however, I have serious concerns about the correlation of experiment and simulations in the present study.

The major point of critique, on which I will focus here, pertains to the quantitation of FG nup surface densities. This is critical to the presented work, because the quantitative comparison of MD simulations with experiments relies on it. The authors chose to use QCM-D frequency shifts to estimate protein surface densities. Yet, in doing so they do not consider that the areal mass densities measured by QCM-D also comprise hydrodynamically coupled solvent in addition to the surface-bound proteins.

The consequences of this error are considerable: NupX surface coverages reported in the manuscript most likely are substantially overestimated. With a reasonable yet conservative estimate of 250 mg/mL protein concentration, the solvent content within the surface-confined NupX layers would be at least 750 mg/mL (to give a total density of at least 1 g/cm³, as would be expected for aqueous protein solutions). This implies that protein surface densities likely are overestimated by at least 4-fold, and dg values likely are underestimated by at least 2-fold.

Specifically, for the conditions shown in Fig. 2 (for NupX films on a gold surface; $\Delta f_5/5 = -28$ Hz) and in Fig. S2 (for NupX films on a silicon nitride surface, as a reference for silicon nitride nanopores; $\Delta f_5/5 = -10$ Hz), the correct dg values would hence be $\geq 2 * 3.5$ nm = 7 nm and $\geq 2 * 5.5$ nm = 11 nm, respectively. These values are comparable to, or even larger than, the Stokes diameter of NupX (7.4 nm according to Table S1). For such low surface coverages, it is difficult to envisage how NupX would form a brush-like planar film, or a meshwork-like pore filling, as obtained in the MD simulations with much large surface densities (Figs. 2J and 4E). In light of these considerations, the seemingly excellent correspondence between experiment and simulations in Fig. 4F would appear coincidental, and one may even ask if the approach used to convert protein concentrations in the pore to conductance is at all meaningful?

I am bemused that the authors fail to acknowledge the substantial contribution of coupled solvent to the areal mass density measured by QCM-D: this phenomenon is well established and has been covered extensively in the literature, including in papers cited by the authors (ref 40, for example, is a seminal review on the analysis of QCM-D data). Several alternative methods to determine protein surface densities have been reported before (including for FG nups). Taken together, I remain to be convinced that the authors have accomplished their goal of reproducing the selective transport barrier of the nuclear pore complex.

Reply to the referees (original comments in black font; our response in blue font; modified text in the manuscript is highlighted in yellow)

Referee #1

This work by Fragasso et al. tests whether it is possible to rationally design a protein capable of recapitulating the selective sorting properties of the “FG” nups. The selective transport mechanism of the nuclear pore complex (NPC) remains a pressing and fundamental problem in cell biology. Through a clear articulation of design parameters and a compelling combination of MD simulations and experimental data, the authors make a strong case for the successful generation of “NupX”. As shown, NupX is capable of binding to the karyopherin Kap95 with an affinity similar to other FG-nups and, most strikingly, exhibits clear selective translocation through NupX-coated nanopores. These data, like all of the data presented in the manuscript, are of excellent quality. Thus, this paper is essentially a proof-of-concept study that opens up new doors to investigate the fundamental properties of the FG-nups that ultimately underlie macromolecular sorting in ways that could be argued are much more controlled than within in vivo settings. It is rather unfortunate, however, that the authors did not attempt to answer any biology-inspired question as the weakness of the work is that it provides little new information about the selective sorting mechanism itself. The one exception here may be the role of so-called spacer regions between the FG repeats that have been suggested by some to impact selectivity; the data presented here suggest that they are not important. A suggestion may be to place this finding in a more prominent position in the paper (e.g. the abstract) to broaden the impact of this work to more biologically minded readers.

We thank the referee for the very positive comments and highlighting the excellent quality of our proof-of-concept study on a rational design of an artificial FG-Nup, and for the suggestions for improvement.

As the referee acknowledges, the main aim of our work was to show a proof-of-concept that designing artificial FG-Nups is a viable approach to studying the size-selectivity mechanism of the NPC. While our current study does not encompass a comprehensive exploration of all factors underlying the size-selectivity of NPCs, we can already infer some important biological insights from the work presented here. As the referee rightly points out, we do observe that the exact amino acid sequence of the spacer regions does not seem to matter for size-selectivity. Instead, *average* spacer properties (such as the polar content, C/H ratio, etc., derived from GLFG-Nups in our case) set the cohesiveness of the protein’s domains and play a governing role.

Following the referee’s suggestion, we now more clearly emphasize this point in the improved manuscript on p. 1 and 3, (abstract and introduction). Moreover, we have (in line with comments from multiple referees) provided density graphs of the two types of FG-motifs in our nanopore simulations (see Figures 4g,h, p. 15). From these, we clearly see that a spatial segregation of FG and GLFG motifs (as reported in Huang *et al.*, Biophys. J. 2020, and Kim *et al.*, Nature 2018), is not required for size-selectivity, although it might play a role in gearing the transport directionality or tuning the transport rates for different types of NTRs. We have mentioned this insight in the manuscript now at multiple points (see p. 1, 3, 16-17 and 20).

Referee #2

In this work, the authors present a strategy for studying nuclear transport, and specifically address the question of whether it is possible to build a synthetic protein that mimics the selective behavior of FG-nups (nucleoporins with FG repeats). They first design a synthetic nucleoporin, NupX, using rational design rules to preserve functionality. As an initial characterization, NupX is compared with other Nups by their amino acid position and content, charge/hydrophobic AA ratios, and disorder scores. Next, they use QCM-D to measure binding affinities of Kap95 (a nuclear transport receptor) to NupX as well as Nsp1 (a native nucleoporin) and Nsp1-s (a mutated nucleoporin) as positive and negative controls, respectively. Additionally, they use molecular dynamics simulations to predict the Kap95/tCherry absorption behavior. After determining that NupX behaves similar to native nucleoporins, they perform single-molecule translocation experiments with NupX-coated nanopores. This aspect of the work is a continuation of “Single-molecule transport across an individual biomimetic nuclear” (Nature Nanotechnology, 2011 Kowalczyk, et al.). BSA is shown to be mainly blocked by the coated pore whereas Kap95 passes through at high event rate. The authors subsequently use coarse-grained molecular dynamics simulations to predict the protein distribution and conductance of a NupX-lined nanopore of different diameters. Lastly, they simulated a NupX-lined nanopore in the presence of both Kap95 and tCherry (control) particles, showing that tCherry is largely excluded whereas Kap95 enters the pore.

The work is very well written, and the results are well-supported by experiment and simulations. In comparison to the former work (referenced above), this one presents sophisticated molecular dynamics simulations, as well as a novel application (i.e. the “first demonstration of a fully artificial nucleoporin”). Design of synthetic nucleoporins and testing in this manner can be used to elucidate the relationship between the FG-nup amino acid sequence and size-selectivity of the NPC.

We thank the referee for the very positive appraisal.

I have listed below a number of key points that could enhance the manuscript, and some suggestions to improve the robustness of the conclusions. I would encourage the authors to consider them in a revision before a recommendation for publication in Nature Communication is made:

- The authors have previously shown tethering of FG-Nups to solid-state nanopores (Kowalczyk et al., Nature Nanotechnology 2011), and low-resolution MD simulations for such systems are routinely published. Therefore, the novelty of this work lies mainly in the development of artificial Nups. However, it is not clear why synthetic Nups such as NupX are needed to study the size selectivity of natural Nups. I would expect this development to have led to additional insights into the interactions between different nucleoporins, or to have highlighted regions of artificial Nups that have surprising or poorly understood functions. It would be useful to comment on this or emphasize the significance of the current study.

We agree that the novelty of our work lies primarily in the development of artificial Nups. We would like to point out that this work provides the first proof-of-concept that a fully artificial intrinsically disordered protein can efficiently mimic the selective behavior of FG-Nups, a feat that is by no means trivial. Our work enables future systematic studies, where systems comprising artificial FG-Nups with well-controlled physiochemical properties can be used to find the essential sequence properties underlying size-selective transport.

A new insight gained through our approach as presented in the paper is that no specific order in the spacers is required to have a selective and functional protein barrier. Moreover, we also show that size-selectivity can be achieved through a homogeneous barrier comprising similarly distributed FG and GLFG

motifs. This indicates that a spatial segregation of different FG-types (as seen in Huang *et al.*, Biophys. J. 2019, and Kim *et al.*, Nature 2018) in itself is not a requirement for size-selectivity. Following the referee's suggestion, we have emphasized the significance of these findings now in Figures 4g,h, and on p. 1, 3, 16-17 and 20.

- Using solid-state nanopores is a nice way to simulate a pore complex in a controlled way, but to what extent is this an accurate model system for natural Nups in an NPC? The conclusion states that "(...) our artificial Nups can be used to build in vitro replicas of the NPC that are suitable for disentangling the NPC transport mechanism", but this is by no means obvious. I would imagine the interaction between different Nups determines their collective transport behavior; it is not clear how the results shown here can be extended to simulate that. How would one go about functionalizing a solid-state pore with several different synthetic Nups in a way that accurately represents NPCs?

We thank the referee for this question. Indeed, the solid-state nanopores outlined here and in earlier work (Jovanovic-Talisman *et al.* 2009, Kowalczyk *et al.* 2011, Ananth *et al.* 2018) only comprise a single type of FG-Nup, whereas even minimally functional native NPCs operate under a heterogeneous set of FG-Nups (i.e., Adams *et al.* Genes Genomes Genetics, 2016). Since all of the SiN-based biomimetic nanopores were shown to be size-selective, this indicates that studying a single type of FG-Nup in an environment that is geometrically similar (30-50 nm diameter for SiN as compared to ~38 nm in yeast) to native NPCs should suffice to elucidate size-selectivity mechanisms. However, the referee is correct in pointing out that solid-state pores do not allow for including interactions between different types of FG-Nups in a controlled way. For this, we recently developed DNA-origami scaffolds (Ketterer *et al.* Nat. Comms 2018) that can comprise different FG-Nups, since the grafting of different proteins to the interior of these scaffolds can be precisely controlled. This approach can be combined with our design protocol for artificial FG-Nups, opening the possibility to systematically study how the interplay between different (artificial) Nups gives rise to various transport properties of the NPC, including size-selectivity. We have more accurately outlined this now on page 20.

- The MD simulations show selectivity for Kap95, and translocations occur without an applied electric field. How does this compare with translocations through the nuclear membrane, and translocations under an external electric field in a solid-state pore? Will the orientation and organization of NupX be affected by an external electric field?

The referee is correct that the MD simulations and in vivo experiments indeed do not have a bias potential, whereas the experiments on biomimetic pores rely on the presence of a voltage bias in order to record translocation events via a conductance blockade. However, as we have showed before (Kowalczyk *et al.* Nat. Nanotech. 2011, Ananth *et al.* eLife 2018), the effect of the applied bias appears to be negligible: From the electrical characterization of NupX-coated pores (see IV curve in Fig. S.7a) we found that the current-voltage characteristics are strictly linear, which means that the NupX-coated pores behave ohmic within the probed range of bias voltages ($\pm 200\text{mV}$), indicating that the NupX-meshwork is not affected appreciably. We now added this additional explanation to p. 11.

- Despite their relative simplicity (single-aa resolution), the MD simulations qualitatively predict the selective transport of Kap95 through a NupX-coated pore. However, they do not provide much insight beyond the experimental results, and therefore the utility of such simulations is questionable for more complex NPCs. I wonder if there is more to be learned from the extensive MD simulations?

The MD simulations provide essential microscopic insight that cannot be obtained from the experiments. For example, from an analysis of the localization of FG and GLFG motifs in our simulations, we learn that the spatial segregation of FG and GLFG motifs observed in other work (Huang *et al.* Biophys

J. 2020 and Kim *et al.* Nature 2018) does not seem to be a requirement for size-selectivity. In addition, the MD simulations reveal a striking rearrangement of FG Nups as a function of pore diameter: for <25 nm nanopores the cohesive domains are largely expelled from the pore region, while for 60 nm nanopores a central opening exists. These MD results provide mechanistic insight on the size-selectivity of the biomimetic nanopores. The similarity in the average Stokes' radius (Fig. 1h) among different NupX designs supports the notion that no specific spacer sequence is required, and that our design method is robust against permutations of the spacer sequence. The excellent correspondence between simulations and experiments in this work shows that the model is able to capture the essential features of these minimal NPCs, which is not trivial. This indicates that our simulations have predictive value and can be applied to more complex systems comprising designer FG-Nups with varying physical properties, thus guiding future experiments. We have clarified the above points on pages 1, 3, 16-17 and 20.

- The MD simulations show that the size selectivity is strongly dependent on the diameter of the pore. In this context, an experimental confirmation for one other pore size (at least) is missing.

We thank the referee for the suggestion to include data for another pore size. We now included data with translocations for both BSA and Kap95 through a 60 nm NupX-coated pore (p. 19 in the SI). We find, as expected, that the pore is selective with an increased event rate for BSA through this larger pore as compared to the 30-35 nm pore sizes. Moreover, we observe faster translocations (~1.6 ms, vs ~4.2 ms for a 30 nm pore) of the BSA molecule through these large nanopores. Both findings can be understood from the simulated density distribution that shows the presence of an empty central region for the 60 nm pore (cf. previous point). We also attempted experiments for smaller pore diameters (<25 nm), but unfortunately could not detect well-resolved translocation events due to the poor signal-to noise-ratio at such small conductances. We now added text on p. 13 to report on these experiments.

- It seems like all the experimental results of translocations are taken from a single nanopore. Given the reliance on these results and, no doubt, the pore-to-pore variation in coating with NupX, I would encourage the authors to include some more statistics on these translocations.

We have reproduced all essential features reported in the paper for multiple pores. We now included scatter plots showing translocation events through a 30 nm, 35 nm and 60 nm NupX-coated pore in the SI (Figure S8, p.19 in the SI). Event rates reported in Fig. 3g were averaged over N=3 different nanopores, with pore diameters in the range 30-35 nm. We added a clarifying statement on this on p. 13.

Minor comments:

- The characterization of binding kinetics (Fig. 2) can be moved to the SI.

We thank the referee for the suggestion. We however think that the data in Fig. 2 deserve to be within the main text since they show selective binding of Kap95 to the synthetic NupX, which we find relevant on its own.

- Define NTR in introduction - nuclear transport receptor
- Define QCM-D in introduction- quartz crystal microbalance with dissipation monitoring

We have added both definitions to the introduction section on p. 3.

- “Whereas PONDR predicts one short folded segment between residues 189 and 209”- perhaps state also the normalized AA position (~0.65) as its presented in Figure 1g.

In referring to Figure 1g, we have added the normalized position (p. 7).

- “assuming an equilateral triangle lattice” – where does this assumption come from, which is also used in the molecular simulation?

We assumed the densest packing possible. A fully triangulated lattice is close-packed in two dimensions (also known as hexagonal packing), i.e. all anchor sites are at equal distance to each other, resulting in one unique length scale that determines the grafting density. We have included this explanation on p. 7 and p. 27.

- Figure S2- “estimated of” to “estimated as”
- “Importantly, the data thus show that the” – should be “shows”
- “z-values ranging from 2 tot 10 nm” –typo
- Figure 4f- “Open circles indicate measured the conductance for...” should be “open circles indicate conductance measurements for...”

We have corrected the above minor mistakes (p. 12 (SI), p. 10 and p. 16.), where the sentence mentioned in point 3 has been removed altogether.

Referee #3

The paper by Fragasso et al nicely demonstrates the capacity of using sequence-designed artificial proteins to build biomimetic nanopore with high molecular selectivity. The substantial experimental and simulation results show that the designed NupX supports fast translocation of transport receptor Kap95 while blocking inert BSA. The experimental system has the potential of serving as a platform to study different FG-Nups of the NPC and to test the performance of sequence-controlled artificial polymers (J. Am. Chem. Soc. 2017, 139, 18, 6422-6430). We think the paper deserves to be published. Below are our questions:

We thank the referee for the appreciative comments and recommendation for publication of the manuscript.

1. The QCM experiment seems to suggest that the Nup films are very rigid. Are there any explanations for this observation?

We thank the referee for pointing this out. We did indeed find that NupX films are quite rigid, as deduced from a lower dissipation-to-frequency ratio (DFR, see Figure S4). We attribute the increased rigidity of the NupX films compared to Nsp1 and Nsp1-S to the lower charge-to-hydrophobicity ratio (C/H) of NupX. A similar result was found in Eisele et al 2013 using the same technique, where Nup98 (a human GLFG-Nup with low C/H, similar to NupX) was compared to Nsp1 and Nsp1-S in terms of DFR (Fig. 4 of that paper), thus finding that Nup98 is more rigid than Nsp1 and Nsp1-S. We added text in the caption of Fig. S4 to emphasize this point.

2. What are the novel scientific insights from this work compared to the previous work with similar system setup from the same authors (eLife 2018;7:e31510)?

Our paper provides the first proof-of-concept that a fully artificial protein can mimic the selective behavior of FG-Nups. Importantly, we found that no specific spacer sequence is required for either binding to Kap95 or to build a selective barrier, suggesting instead that overall properties such as C/H, disorder, and the presence of equally spaced FG and GLFG motifs are sufficient to achieve functionality.

Moreover, there is no spatial segregation of FG and GLFG motifs in our NupX-functionalized systems, as observed in simulations of native yeast NPCs (Kim *et al.* Nature 2018, Huang *et al.* Biophys. J. 2020). Thus, our work shows that spatial segregation of different types of FG-motifs is not required for size-selectivity (although its effect on other transport functions such as directionality and transport rate cannot be ruled out). We have added these points in Figures 4g and h, and clarified them further on p. 16, 17 and 20, and included these more clearly in the abstract and introduction sections (p. 1 and 3)

3. Our recent paper as cited by the authors (bioRxiv 568865 (2019). doi:10.1101/568865, now in press by the biophysical journal) shows that different FG groups occupy different spatial territories, suggesting that they have distinct/complementary functions. In the NupX designed by the authors, there are both GLFG and FG repeats. Have the authors investigated the different effects of these two FG groups on the structure and function of the FG-Nups?

We thank the referees for highlighting this. Inspired by this question, we now performed an analysis on the localization of FG and GLFG motifs within the nanopore geometries, as was done already for the NupX-proteins in a brush geometry (figure 2l). These densities have been added to Figures 4g and h, together with guiding/clarifying statements on p. 16-17 and p. 20, and are now mentioned in the abstract and the introduction (p. 1 and 3). The spatial segregation, as found by the referees and in Kim *et al.*, Nature 2018, is absent in our size-selective nanopores. Rather, the distributions of FG and GLFG-motifs strongly overlap and correlate well with the overall protein distribution. From these findings, we can conclude that spatial segregation of FG and GLFG motifs is not essential for size-selectivity. This does, however, not rule out other functional roles for spatial segregation in NPCs, such as, for example, transport directionality. In future studies, one could use our approach to design artificial FxFG-type Nups similar to the GLFG-type NupX, or artificial Nups with varying FxFG or GLFG-content and investigate the role of spatial segregation of different motifs and their arrangement on nuclear transport. Moreover, using DNA origami (see Ketterer *et al.*, Nat. Comms. 2018), further control can be obtained over the spatial arrangement of different FG-Nups, implying that spatial segregation of FG motifs can be systematically studied experimentally in future studies. We now clarified these points on p. 20.

4. How did the authors choose the interaction parameter between NupX and Kap95 in their simulations?

We employed a similar approach in modelling Kap95 and the inert control protein tCherry as in recent modelling work (see Ananth *et al.* eLife 2018 and Mishra *et al.* IJMS 2019): We applied a volume-exclusion potential (i.e., only the repulsive term of Φ_{HP}) as the non-bonded potential between NupX amino acid beads and surface beads of the Kap95/tCherry particles. On top of the non-bonded Lennard-Jones/hydrophobic potentials, the electrostatic interactions between charged amino acids and the partial charges on the surface of the Kap95 particle follow the modified Coulomb potential (Φ_{EL}). The binding sites of the Kap95 particle are modelled as hydrophobic particles with a hydrophobicity and size equal to a Phe-bead within the 1-BPA forcefield. We have added a clarification on the interactions between amino acid beads and Kap95/tCherry in the Materials and Methods section on p. 27 and 28.

Referee #4

The authors set out to fabricate synthetic pores that reproduce the selectivity of macromolecular binding and permeation in the nuclear pore complex (NPC). This is a challenging goal which, if successfully accomplished, should be of interest to a broad readership in the nanosciences, bottom-up synthetic biology, and possibly other research areas.

The manuscript is based on a novel method for the design of artificial FG nups. The rationale behind the method, described in the first part of the Results section, appears sensible overall. Whilst the exact method is a matter of taste to some extent in light of the various physical models of selective permeability that have been proposed, it is a good starting point for bottom-up reconstitution efforts.

The authors' approach to combine data from two distinct geometries (i.e. FG nups grafted either to a planar surface, or to the inner wall of a nanopore) and to correlate experimental data with coarse-grained molecular dynamics (MD) simulations for each of these two geometries, has its merit: The nanoscale analysis of assemblies of intrinsically disordered proteins remains technically difficult, and the combination of experiment and theory is a promising approach to tackle this challenge as already shown by previous work in the field.

We thank the referee for the positive appraisal.

Unfortunately, however, I have serious concerns about the correlation of experiment and simulations in the present study. The major point of critique, on which I will focus here, pertains to the quantitation of FG nup surface densities. This is critical to the presented work, because the quantitative comparison of MD simulations with experiments relies on it. The authors chose to use QCM-D frequency shifts to estimate protein surface densities. Yet, in doing so they do not consider that the areal mass densities measured by QCM-D also comprise hydrodynamically coupled solvent in addition to the surface-bound proteins.

We thank the referee for expressing the need for caution on the interpretation of our QCM-D data, especially considering the presence of solvent. Below we will address these technical concerns.

The consequences of this error are considerable: NupX surface coverages reported in the manuscript most likely are substantially overestimated. With a reasonable yet conservative estimate of 250 mg/mL protein concentration, the solvent content within the surface-confined NupX layers would be at least 750 mg/mL (to give a total density of at least 1 g/cm³, as would be expected for aqueous protein solutions). This implies that protein surface densities likely are overestimated by at least 4-fold, and dg values likely are underestimated by at least 2-fold.

Specifically, for the conditions shown in Fig. 2 (for NupX films on a gold surface; $\Delta f_5/5 = -28$ Hz) and in Fig. S2 (for NupX films on a silicon nitride surface, as a reference for silicon nitride nanopores; $\Delta f_5/5 = -10$ Hz), the correct dg values would hence be $\geq 2 * 3.5$ nm = 7 nm and $\geq 2 * 5.5$ nm = 11 nm, respectively. These values are comparable to, or even larger than, the Stokes diameter of NupX (7.4 nm according to Table S1). For such low surface coverages, it is difficult to envisage how NupX would form a brush-like planar film, or a meshwork-like pore filling, as obtained in the MD simulations with much large surface densities (Figs. 2J and 4E). In light of these considerations, the seemingly excellent correspondence between experiment and simulations in Fig. 4F would appear coincidental, and one may even ask if the approach used to convert protein concentrations in the pore to conductance is at all meaningful?

I am bemused that the authors fail to acknowledge the substantial contribution of coupled solvent to the areal mass density measured by QCM-D: this phenomenon is well established and has been covered extensively in the literature, including in papers cited by the authors (ref 40, for example, is a seminal review on the analysis of QCM-D data). Several alternative methods to determine protein surface densities have been reported before (including for FG nups). Taken together, I remain to be convinced that the authors have accomplished their goal of reproducing the selective transport barrier of the nuclear pore complex.

We thank the referee for his/her strong endorsement of the overall approach of our paper, but also for pointing us to the important aspects of converting the QCM-D data to brush densities. In our revised manuscript, we now provide a more accurate estimation of the protein grafting density, see Section 11 of the SI and Materials and Methods, p. 25. We find a very similar average grafting distance for NupX layers on SiN but higher values on Au. For the improved estimate, we used the Voigt-Voinova viscoelastic model (explained in Dutta *et al.*, Langmuir, 2007) to fit our QCM-D data and estimate the Voigt mass, which accounts for both the viscoelastic behavior and the wet mass density of the film (protein + coupled water). With a 0.35 fraction (from a recent review from Huang et al. 2017) for the effective protein mass and 0.65 for the coupled water, we calculated the areal mass densities for all the proteins. This yielded average grafting distances of 6.2 nm for NupX on SiN, 5.6 nm for NupX on gold, 5.8 nm for Nsp1 on gold, and 5.4 nm for Nsp1-S on gold.

In view of the significantly updated value (5.6 nm vs 3.5 nm) for the grafting distance found for NupX on Au, we re-did our MD simulations on the absorption of Kap95 and tCherry-sized particles into NupX brushes using the newly estimated grafting distance. In the updated Fig. 2I, we now show new simulation snapshots, the time-averaged density profile, and the PMF curves associated with the absorption of Kap95 and a tCherry-sized particle, for a NupX-brush with a grafting distance of 5.7 nm. We find that the density profile of the protein brush has become more uniform, with an average density of 170 mg/mL throughout the body of the brush. At the same time, and most importantly, the NupX brush still shows a preferential absorption of Kap95 over a tCherry-sized particle. Gratifyingly, the K_D estimated from the free energy profile is now on the order of $\sim\mu\text{M}$, indicating an even better correspondence of our simulation result to the experimentally measured K_D -value ($10^{-1} \mu\text{M}$). We have discussed these new findings on p. 10 in the revised manuscript.

For the SiN nanopores, the new analysis revealed only a small change of 10% in grafting distance (6.2 nm vs 5.5 nm). We performed a sensitivity analysis by performing simulations on the nanopore and found that a change in grafting distance from 5.5 nm to 6 nm ($\sim 10\%$ increase with respect to the original value of 5.5 nm) reduced the protein density within the nanopores by only $\sim 15\%$, with no qualitative changes in its distribution, see Fig. S12. From this we conclude that the results presented in Figs. 4 and 5 are solid.

We would like to thank the referee for this very useful feedback, which helped to improve the quality of our work.

Reviewer #2 (Remarks to the Author):

The revised manuscript fully addresses my comments. The overall impact and clarity of the paper has been greatly improved with the additional data and refined interpretations. I recommend publication in Nature Communications.

Reviewer #3 (Remarks to the Author):

We found the revised manuscript improved, but unfortunately we cannot recommend the publication of the paper in its current form because of the following problems.

We are not convinced by the two new statements below, which could be over-concluding and misleading.

1. In the abstract, “and demonstrate that no specific spacer sequence nor a spatial segregation of different FG-motif types are needed to create functional NPCs.”

2. In page 20, “we are confident that a single artificial FG-Nup can capture the complex transport behaviour of the NPC.”

The authors reduced the function of the NPC to mere cargo size selectivity. One needs to be reminded that the NPC simultaneously controls the transport of many different cargoes with both high selectivity and high efficiency. We think there are still many open questions regarding the complex transport properties of the NPC and therefore a large gap between the presented simple artificial system and the NPC.

The interpretation of the QCM-D results is still not satisfactory. There are four unknowns regarding the properties of the water-rich Nup film: the density, the thickness, and the complex shear modulus. It is not clear to us how the pair of frequency and dissipation shifts provided by QCM-D are enough to solve the four unknowns, not to mention that the property of the film could vary at different heights and a sharp line might not exist between the film and the bulk water to define the film thickness.

Reviewer #4 (Remarks to the Author):

I appreciate the authors' effort to extract protein surface densities from QCM-D data. This is challenging, as has been pointed out in many papers over the last two decades (including the previously mentioned review, now ref. 41). The contribution of hydrodynamically coupled solvent to the areal mass density as measured by QCM-D can vary over a wide range, from as little as 10% (e.g. for supported lipid bilayers) to more than 98% (e.g. for swollen polymer films), and depends sensitively on the organisation of biomolecules on the surface. In this context, it is not obvious why the value of 65% reported by Huang et al (ref. 43) for a set of three globular proteins should be applicable to intrinsically disordered nucleoporins.

The following simple check shall illustrate that there is a serious inconsistency in the analysis

proposed by the authors. I consider the QCM-D data for NupX (Figure 2a and Table S4), and the authors' assumption that proteins account for 35% (and thus solvent for 65%) of the total areal mass density measured by QCM-D. The partial specific volume of proteins is 0.73 ml/g to a good approximation (Zhao et al Biophys J 2011, 100:2309) and the density of the aqueous solvent is 1.0 g/mL. With these numbers, one can calculate a film density of 1.1 g/mL, and a protein concentration in the film of $0.35 \times 1.1 \text{ g/mL} = 390 \text{ mg/mL}$. Moreover, the total areal mass density given in Table S4 corresponds to a film thickness of $556 \text{ ng/cm}^2 / (1.1 \text{ g/cm}^3) = 5.1 \text{ nm}$. These values can be compared to the simulation data in Fig. 2l which was obtained with nominally very similar grafting densities.

It is clear that the protein concentration based on the assumption of 35% mass contribution (390 mg/mL) exceeds the concentrations predicted by the simulations substantially. For example, considering the plateau (170 mg/mL) as a representative concentration, the discrepancy is larger than a factor of 2! An equivalent discrepancy can be observed for the thickness. I note here that the comparison is not trivial because the models typically used to analyse the QCM-D data assume a homogeneous film whereas the simulations reveal that the film density decreases gradually with the distance from the surface. However, it is well established that QCM remains rather sensitive even in the interfacial region between film and bulk solution where the protein concentration is low (see Domack et al Phys Rev E 1997, 56:680, for example). The QCM thickness of a films as shown in Fig. 2l would likely be around 15 nm, or about 3 times larger than measured in the experiment.

I trust that this simple example demonstrates the analysis provided by the authors is inconsistent. A possible explanation is that the experimental grafting densities remain substantially overestimated. Another possible explanation is that the simulations do not adequately reproduce the real properties of surface-anchored NupX.

In conclusion, I remain unconvinced that nucleoporin surface densities have been determined properly or that the manuscript demonstrates a good match between experiment and simulations. The authors may use alternative methods that are better suited to quantify areal mass densities if they wish to resolve these issue. Optical waveguide lightmode spectroscopy or spectroscopic ellipsometry, for example, are well suited for the purpose and now routinely available in a number of laboratories.

This is only one of my concerns related to the manuscript, but is sufficient to illustrate the lack of rigour in the analysis. Other examples can be given; however, I consider it the task of the authors to provide proper evidence for all conclusions made rather than relying on reviewers as proof readers.

Reply to the referees (original comments in black font; our response in blue font)

Reviewer #2 (Remarks to the Author):

The revised manuscript fully addresses my comments. The overall impact and clarity of the paper has been greatly improved with the additional data and refined interpretations. I recommend publication in Nature Communications.

We thank the reviewer for the constructive comments and for recommending our paper for publication in Nature Communication.

Reviewer #3 (Remarks to the Author):

We found the revised manuscript improved, but unfortunately we cannot recommend the publication of the paper in its current form because of the following problems.

We are not convinced by the two new statements below, which could be over-concluding and misleading.

1. In the abstract, “and demonstrate that no specific spacer sequence nor a spatial segregation of different FG-motif types are needed to create functional NPCs.”
2. In page 20, “we are confident that a single artificial FG-Nup can capture the complex transport behaviour of the NPC.”

The authors reduced the function of the NPC to mere cargo size selectivity. One needs to be reminded that the NPC simultaneously controls the transport of many different cargoes with both high selectivity and high efficiency. We think there are still many open questions regarding the complex transport properties of the NPC and therefore a large gap between the presented simple artificial system and the NPC.

We agree and thank the reviewer for these suggestions. We rephrased text in the abstract and on now p.13 accordingly.

The interpretation of the QCM-D results is still not satisfactory. There are four unknowns regarding the properties of the water-rich Nup film: the density, the thickness, and the complex shear modulus. It is not clear to us how the pair of frequency and dissipation shifts provided by QCM-D are enough to solve the four unknowns, not to mention that the property of the film could vary at different heights and a sharp line might not exist between the film and the bulk water to define the film thickness.

We appreciate the question of the reviewer on the QCM-D data analysis. Our intention in including the Voigt-Voinova analysis was to create a more reliable estimate of the grafting distance than the standard Sauerbrey analysis. We have now expanded our data set and independently characterized the grafting density of our proteins using surface plasmon resonance (SPR), and furthermore performed a new set of QCM-D experiments under matching incubation conditions. The SPR data provide a direct estimation of the dry mass of our protein layers, from which we calculated the grafting distance. The new SPR data forego the need to perform the Voigt-Voinova analysis, and hence the latter is not considered anymore in the newest version of the manuscript. For a summary of the additional steps that we took in determining the grafting distance of our NupX proteins, we refer to our response to reviewer #4 below.

Reviewer #4 (Remarks to the Author):

I appreciate the authors' effort to extract protein surface densities from QCM-D data. This is challenging, as has been pointed out in many papers over the last two decades (including the previously mentioned review, now ref. 41). The contribution of hydrodynamically coupled solvent to the areal mass density as measured by QCM-D can vary over a wide range, from as little as 10% (e.g. for supported lipid bilayers) to more than 98% (e.g. for swollen polymer films), and depends sensitively on the organisation of biomolecules on the surface. In this context, it is not obvious why the value of 65% reported by Huang et al (ref. 43) for a set of three globular proteins should be applicable to intrinsically disordered nucleoporins.

The following simple check shall illustrate that there is a serious inconsistency in the analysis proposed by the authors. I consider the QCM-D data for NupX (Figure 2a and Table S4), and the authors' assumption that proteins account for 35% (and thus solvent for 65%) of the total areal mass density measured by QCM-D. The partial specific volume of proteins is 0.73 ml/g to a good approximation (Zhao et al Biophys J 2011, 100:2309) and the density of the aqueous solvent is 1.0 g/mL. With these numbers, one can calculate a film density of 1.1 g/mL, and a protein concentration in the film of $0.35 \times 1.1 \text{ g/mL} = 390 \text{ mg/mL}$. Moreover, the total areal mass density given in Table S4 corresponds to a film thickness of $556 \text{ ng/cm}^2 / (1.1 \text{ g/cm}^3) = 5.1 \text{ nm}$. These values can be compared to the simulation data in Fig. 2l which was obtained with nominally very similar grafting densities.

It is clear that the protein concentration based on the assumption of 35% mass contribution (390 mg/mL) exceeds the concentrations predicted by the simulations substantially. For example, considering the plateau (170 mg/mL) as a representative concentration, the discrepancy is larger than a factor of 2! An equivalent discrepancy can be observed for the thickness. I note here that the comparison is not trivial because the models typically used to analyse the QCM-D data assume a homogeneous film whereas the simulations reveal that the film density decreases gradually with the distance from the surface. However, it is well established that QCM remains rather sensitive even in the interfacial region between film and bulk solution where the protein concentration is low (see Domack et al Phys Rev E 1997, 56:680, for example). The QCM thickness of a films as shown in Fig. 2l would likely be around 15 nm, or about 3 times larger than measured in the experiment.

I trust that this simple example demonstrates the analysis provided by the authors is inconsistent. A possible explanation is that the experimental grafting densities remain substantially overestimated. Another possible explanation is that the simulations do not adequately reproduce the real properties of surface-anchored NupX.

We thank the reviewer for pointing out these uncertainties in the estimated grafting distances of our proteins. Further below we indicate the steps that we took in order to rigorously determine the grafting distance of our NupX-functionalized surfaces, which include additional QCM-D measurements and measurements using the surface plasmon resonance (SPR) technique.

In conclusion, I remain unconvinced that nucleoporin surface densities have been determined properly or that the manuscript demonstrates a good match between experiment and simulations. The authors may use alternative methods that are better suited to quantify areal mass densities if they wish to resolve these issue. Optical waveguide lightmode spectroscopy or spectroscopic ellipsometry, for example, are well suited for the purpose and now routinely available in a number of laboratories. This is only one of my concerns related to the manuscript, but is sufficient to illustrate the lack of rigour in the analysis. Other examples can be given; however, I consider it the task of the authors to provide proper evidence for all conclusions made rather than relying on reviewers as proof readers.

We have followed the specific suggestion of the reviewer to employ a separate technique to determine the grafting densities of our Nup-coated surfaces. We decided to use an optical technique, namely Surface Plasmonic Resonance (SPR), to estimate the dry mass of our proteins bound to both gold and silica substrates. These measurements were performed by an external expert on the SPR technique, prof. Andreas Dahlin from Chalmers, who has broad experience with SPR analysis of a variety of Nups. This led to the involvement of two additional authors on this manuscript.

We performed SPR measurements of different surface configurations of Nups, both to determine the grafting distance of NupX on all employed surfaces (SiN and Au), and to show that the chemistry used to functionalize the surfaces worked as intended. These measurements led to major revisions of the manuscript, namely text revisions on p.5-7, inclusion of an updated Fig. 2 on p.25, and an inclusion and description of SPR data in Fig. S6 on p.15 of the SI.

To be specific, we performed the following additional measurements and analyses:

- Thickness measurement of silica-coated SPR chips incubated with 1 μ M NupX
- Thickness measurement of Au-coated SPR chips incubated with varying concentrations of NupX (60nM and 2 μ M)
- Thickness measurement of Au-coated SPR chips incubated with varying concentrations of MUTEG (1.3 μ M to 1.3 mM)
- Thickness measurement of APTES/Sulfo-SMCC-functionalized silica surfaces
- QCM-D measurements of NupX incubation on Au under conditions similar to those used to coat the SPR chips. In this way we ensured that the grafting distances obtained from SPR were representative for those in the QCM-D measurements
- QCM-D measurements of Kap95/BSA flushes on the above mentioned NupX-coated Au chips.

From these additional measurements, we found that:

- (1) Coating of silica-coated SPR chips with 1 μ M NupX for \sim 1 hr (including silanization) yielded a grafting distance of 5.4 ± 1.1 nm, which corresponds very well to the value used as input value for the MD simulations of the biomimetic nanopore (5.5 nm). The underlying SPR data and grafting distance estimation are now reported in SI p.15 and Fig. S6. The coating protocol of the SPR chips, data acquisition and instrumentation are reported in the 'Materials and Methods' section on p.17.
- (2) Coating of gold-coated SPR chips with NupX for \sim 1 hr yielded a grafting distance of 7.7 nm for 60nM, and 2.9 nm for 2 μ M, showing that we can tune the grafting distance by varying the protein concentration in the incubation solution. Importantly, we repeated our QCM-D experiments using different concentrations of NupX (100 nM, 500 nM, 1 μ M, 2 μ M), such that the incubation conditions are similar between the SPR measurements and the QCM-D results. Indeed, we found frequency shifts after \sim 1hr of incubation that vary with the NupX concentration, consistent with our SPR findings. We then showed that Kap95 still binds to NupX layers under all these incubation conditions, whereas BSA does not. The underlying SPR data and estimation are now reported in SI p.15. QCM-D data are now illustrated in a revised version of Fig. 2 and Fig. S4.
- (3) Coating of gold-coated SPR chips with Nsp1 and Nsp1-S provided grafting distances of about 4.8 nm and 5.8 nm, respectively. SPR data are reported in SI p.15.
- (4) The MUTEG passivation step, which is used to prevent unwanted association of BSA or Kap95 to the residual bare Au surface in our QCM-D experiments, does form monolayers with a measured grafting distance of \sim 0.8 nm for the 1.3 mM incubation concentration that is also employed in passivating the NupX layers before our binding affinity experiments.

As indicated in the response to reviewer #3, our intention behind the Voigt-Voinova analysis was to estimate the grafting distance in a more reliable way than in the standard Sauerbrey analysis. Given the strongly expanded data set with the independent SPR measurements that we now added, however, we now decided to not include this analysis anymore in the revised version of our manuscript. The extensive characterization of our surfaces using SPR in addition to new QCM-D measurements indicates that dense NupX layers are formed. The absorption results from our QCM-D experiments indicate selective binding over a range of grafting distances: for sparse NupX brushes with grafting distances as high as 7.7 nm, down to densely grafted NupX brushes with a grafting distance of 2.9 nm.

Based on our SPR measurements on both NupX-coated Au and SiN chips, we find that the grafting distances used in our MD simulations and our *in vitro* experiments are in good agreement: the values used for our brush simulations (4.0 nm and 5.7 nm) fall well in the range of values for which we demonstrate selective adsorption (2.9 - 7.7 nm), and the value used in our nanopore simulations (5.5 nm) is close to the value obtained for SiN surfaces (5.4 nm).

We thank the reviewer for the useful feedback, which led us to improve the rigor of our data analysis.

Reviewer #3 (Remarks to the Author):

We found the revised manuscript improved and ready for publication.

Reviewer #4 (Remarks to the Author):

See attached file.

The authors have provided new and essential data that resolve some of the major questions I had previously raised. However, several substantial comments still remain. Without these being addressed, I remain unconvinced that the correlation between experiment and theory, and between FG Nups as planar films and pores, is solid.

In addition, I also provide a rather long list of additional minor comments. I trust the authors can readily address these to improve the clarity of the data and the manuscript.

Major comments:

1. Fig. S5: The authors state that MUTEg is expected to form a 2 nm thick layer. Whilst I agree with that statements based on the molecular structure, I note that Fig. S5 suggests a very different result. A frequency shift of -40 Hz after washing corresponds to a thickness of at least 7 nm. The dissipation shift is also quite high, again inconsistent with a dense 2 nm thick film. Moreover, the $\Delta D/\Delta f$ ratio is quite high ($0.3 \times 10^{-6}/\text{Hz}$), implying the film might be substantially thicker than the 7 nm estimated based on the Sauerbrey equation. Whilst the MUTEg organisation may be different when incubated onto a pre-formed FG Nup film (as compared to incubation on plain gold), the data shown in Fig. S5 raise a serious concern: does MUTEg have a major effect on the organisation (and ultimately, function) of the FG Nup film? This has implications for the comparison between experiment and simulation on planar FG Nup films, and for the comparison between data for planar surfaces versus pores. Moreover, it also raises the question how meaningful the effective distance between MUTEg molecules, as determined by SPR, is. Surely, it is unlikely to be equivalent to the grafting density (as the authors claim) because not all molecules can be grafting to the gold in a MUTEg film that in its solvated state is (at least) 7 nm thick.
2. Figs. 2a, d, g, S2, S3 and S5a: What was the composition of the solutions used before, during and after FG Nup and MUTEg incubations in QCM-D experiments? This is critical to be able to understand the data, but I could not find this information. Related to this, can the authors correlate FG Nup film thickness in the experiment and in the simulations as a consistency check? An approximate analysis of the QCM-D data using the Sauerbrey equation to extract experimental film thicknesses should be straightforward, and can be compared with the density profiles obtained from simulations.
3. Page 6: The authors refer to Kap95 release upon washing in the main text but these data are completely lacking (for Nsp1), or shown for just a few minutes (for NupX) in Fig. 2. The unbinding behaviour is as important as the binding behaviour to assess the characteristics of the interaction, and should be shown. Enough data should be presented for the reader to be able to appreciate how slow unbinding is, and if there is a significant fraction of stably bound material. I note in passing that the authors should also show the results of washing in 0.2 M NaOH, rather than indicating 'data not shown' (a supplementary figure is sufficient).

4. SPR data analysis looks solid. However, I note that SPR data were acquired on dried films prior to MUTEg passivation. How did the authors ascertain the data thus obtained are representative for the FG Nups films after MUTEg passivation? It appears quite possible that a proportion of FG Nups is strongly physisorbed (and not bound via a thiol) to the gold prior to MUTEg passivation (and thus included in the SPR measure) but becomes displaced upon MUTEg passivation (and thus is not present in subsequent protein binding assays). Also, I notice that the surface was washed in pure water and ethanol prior to drying. Did the authors check if this affects the amount of bound FG Nups?

Minor comments:

1. Page 2: "As is evident from the multitude of transport models, no consensus on the NPC transport mechanisms has yet been reached." – the multiplicity of models *per se* is not an indicator of a lack of consensus (or more generally, of the state of this research area), in particular considering that the references provided cover almost 2 decades of research.
2. Page 6: The authors use a two-component Langmuir isotherm to quantify binding strength. The authors should clarify the meaning of the obtained values to avoid misleading interpretation by lay readers. Considering the characteristics of the interacting molecules (*i.e.*, their multivalency), an argument can be made that it is unlikely the two K_d values represent two well-defined yet distinct types of binding sites within the FG nup films. Rather, the obtained values should be considered 'effective' and may represent a wide (and Kap95 concentration dependent) spectrum of binding sites and strengths. This was already discussed in previous work by others, and should be acknowledged.
Independent from the above, yet related to it, I note that the authors implicitly assume that frequency shifts are proportional to adsorbed amounts of Kap95. More often than not, this is not the case (for examples, see Reviakine et al Anal Chem 2011, 83:8838, and references therein). This is another reason that the determined K_d values should be considered effective values, and the authors should acknowledge this.
3. The authors should point out that NupX is substantially shorter than any of the native GLFG domains.
4. Page 7: The detailed description of the FG nup density profile refers to a 4.0 nm anchor spacing. This could be made clearer in the text.
5. Page 7: Kap95 was modelled as a sphere matching the total charge density of the protein. Is this meaningful, *i.e.* are all charged amino acids in Kap95 exposed on the protein surface?
6. Page 8: The authors attribute transient dips in current to single molecule translocation. This merits further comment. Do the authors propose protein translocation to be 'single file'? This would be surprising given that an NPC is thought to contain many nuclear transport receptors at any given time (as the authors acknowledge in the introduction). Based on their affinity measurements for NupX and literature data for other FG nups, the authors should be able to estimate the equilibrium amount of Kap95 expected to be located within the pore. How does the result compare with single molecule translocation?
7. Page 9: Can the authors comment on why the blockade times in NupX functionalised pores are longer for BSA than for Kap95?
8. Page 10: "Note that we adopted a critical protein density of 85 mg/ml from the earlier work on Nsp1³¹ while employing a different dependency of the conductivity on the local protein density (Materials and Methods)." The implications are difficult to understand in the context of the main section. Can the authors clarify, in a simple way, how the seemingly excellent match between experiments and simulations shown in Fig. 4f should be interpreted?
9. Page 11: The authors provide an estimate of a binding energy of just 3 k_bT . This appears rather low compared to the effective affinities determined experimentally on planar NupX films. Can the authors comment on this discrepancy?
10. Page 16: The meaning of 'increase in viscoelasticity' is not clear.
11. Page 17: The authors stated QCM-D data were filtered. This is unusual; please clarify what type of treatments the original data were subjected to.
12. Fig. 1b, what does the pink colour for amino acids represent?

13. Fig. 2: Details in the dissipation response are difficult to see. Can the authors adjust the scale to facilitate a clearer view of the data? Ditto for all QCM-D data shown in the manuscript and supporting information.
14. Fig. 4b, g, h: The densities appear asymmetric along the axis. Can the authors comment on this?
15. Supplementary figures are not numbered by order of appearance in the main text. Moreover, some supplementary figures are not even mentioned in the main text (*e.g.*, Fig. S7). This makes reading unnecessarily cumbersome.
16. Fig. S1: Can the authors also show a gel demonstrating the purity of the Kap95 prep.
17. Fig. S2: The authors claim the frequency increase upon washing is due to release of proteins. Another common source of rapid shift upon washing is differences in the solution viscosity or density. How did the authors discriminate between these two scenarios?
18. Fig. S3: The authors claim the frequency shift for MUTEg binding is inversely proportional to NupX binding. The authors will want to demonstrate this is (quantitatively) true, or revise the claim for clarity.
19. Fig. S7: This plot is potentially very informative, but essential information is missing to assess this in full. Are the data shown for FG Nup incubation alone, or do they include the MUTEg passivation step? The latter would be more informative because this reflects the properties of the film used for protein binding studies.
Also, I am surprised the ratios of dissipation shifts over frequency shifts are positive. They should be negative.

Reply to the referees (original comments in black font; our response in blue font)

The authors have provided new and essential data that resolve some of the major questions I had previously raised.

It is good to see that the reviewer acknowledges the value of the new data that we added, as they resolved major questions that he/she previously raised.

However, several substantial comments still remain. Without these being addressed, I remain unconvinced that the correlation between experiment and theory, and between FG Nups as planar films and pores, is solid.

In addition, I also provide a rather long list of additional minor comments. I trust the authors can readily address these to improve the clarity of the data and the manuscript.

Major comments:

1. Fig. S5: The authors state that MUTEg is expected to form a 2 nm thick layer. Whilst I agree with that statements based on the molecular structure, I note that Fig. S5 suggests a very different result. A frequency shift of -40 Hz after washing corresponds to a thickness of at least 7 nm. The dissipation shift is also quite high, again inconsistent with a dense 2 nm thick film. Moreover, the $\Delta D/\Delta f$ ratio is quite high ($0.3 \times 10^{-6}/\text{Hz}$), implying the film might be substantially thicker than the 7 nm estimated based on the Sauerbrey equation.

Whilst the MUTEg organisation may be different when incubated onto a pre-formed FG Nup film (as compared to incubation on plain gold), the data shown in Fig. S5 raise a serious concern: does MUTEg have a major effect on the organisation (and ultimately, function) of the FG Nup film? This has implications for the comparison between experiment and simulation on planar FG Nup films, and for the comparison between data for planar surfaces versus pores.

Moreover, it also raises the question how meaningful the effective distance between MUTEg molecules, as determined by SPR, is. Surely, it is unlikely to be equivalent to the grafting density (as the authors claim) because not all molecules can be grafting to the gold in a MUTEg film that in its solvated state is (at least) 7 nm thick.

We thank the reviewer for bringing up this point that prompted us to thoroughly re-address the MUTEg control data.

We repeated the MUTEg passivation experiments on both bare gold and Nup-coated gold with a fresh batch of MUTEg and reproducibly found final frequency shifts ranging from 10-14 Hz, corresponding to thicknesses of ~1.8-2.3 nm, consistent with previously reported values for MUTEg monolayers (Pale-Grosdemange et al. J. Am. Chem. Soc. 1991). Subsequently, we repeated all experiments involving MUTEg and found that:

- Passivation of bare gold and Nup-coated gold with 1mM MUTEg leads to frequency shifts of ~10-14 Hz and close to zero dissipation (new Figs. S4-S5), indicating the formation of a thin monolayer that does not interact with Kap95 (Fig. S5).

- SPR measurements of dry MUTEg films under the same incubation conditions yielded an average thickness of ~ 1.9 nm (Fig. S3b).
- Gold surfaces coated with different NupX incubation concentrations (of 100nM, 1 μ M, and 2 μ M, Fig. S2) and passivated with 1mM of fresh MUTEg (Fig. S4) were found to interact selectively with Kap95 as compared to BSA (Figs. 2b, S7).
- The K_d constants that we obtain for Kap95 interacting with 1 μ M NupX were found to be comparable to our previous findings (Fig. 2c).

These new data are unambiguous and clearly show that our MUTEg passivation produces dense monolayers that are ~ 2 nm in thickness and have a mean grafting distance of 0.59 ± 0.01 nm from SPR (Fig. S3b). These layers efficiently prevent Kap95 absorption to bare gold (Fig. S5b). Additionally, the near-zero final shift in dissipation does not indicate any significant changes occurring in the NupX brushes.

2. Figs. 2a, d, g, S2, S3 and S5a: What was the composition of the solutions used before, during and after FG Nup and MUTEg incubations in QCM-D experiments? This is critical to be able to understand the data, but I could not find this information.

The composition of the solutions for the affinity experiments is now clarified on p. 16 of the manuscript and in Figures S2 and S4 of the SI.

Related to this, can the authors correlate FG Nup film thickness in the experiment and in the simulations as a consistency check? An approximate analysis of the QCM-D data using the Sauerbrey equation to extract experimental film thicknesses should be straightforward, and can be compared with the density profiles obtained from simulations.

We have now added a height estimate based on a Sauerbrey analysis of the QCM-D data and compare this value to estimates extracted from the MD simulations (p. 7). Generally, the simulated brushes are more extended than a Sauerbrey estimate of the experimental brush height suggest, but such a comparison comes with an important caveat that we explain in the main text on p. 7 and reiterate here: The numbers that we provide based on the Sauerbrey-based estimation from Figs. 2 and S2 should be seen as *lower limits*; as was indicated rightly by the reviewer at an earlier point. Given that the dissipation-to-frequency ratio for the NupX brushes (before MUTEg incubation, Fig. S9) can be as high as $\sim 0.45 \times 10^{-7}$ Hz⁻¹, the actual brush height might be substantially higher than the frequency shifts suggest: this is apparent in other studies on FG-Nups as well (Eisele et al. Embo Rep. 2010, Eisele et al. Biomacromolecules 2012).

3. Page 6: The authors refer to Kap95 release upon washing in the main text but these data are completely lacking (for Nsp1), or shown for just a few minutes (for NupX) in Fig. 2. The unbinding behaviour is as important as the binding behaviour to assess the characteristics of the interaction, and should be shown. Enough data should be presented for the reader to be able to appreciate how slow unbinding is, and if there is a significant fraction of stably bound material.

We have extended the data acquisition time window in Fig. 2 for both NupX and Nsp1 to show ~ 30 min of Kap95 dissociation.

I note in passing that the authors should also show the results of washing in 0.2 M NaOH, rather than indicating 'data not shown' (a supplementary figure is sufficient).

We added a QCM-D time trace that demonstrates complete dissociation of Kap95 upon washing with 0.2M NaOH in Fig. S6.

4. SPR data analysis looks solid. However, I note that SPR data were acquired on dried films prior to MUTEg passivation. How did the authors ascertain the data thus obtained are representative for the FG Nup films after MUTEg passivation?

MUTEg passivation is not expected to affect the grafting distance of FG-Nup films, nor is there any evidence in our data that suggest or imply that this might be the case (as detailed below).

It appears quite possible that a proportion of FG Nups is strongly physisorbed (and not bound via a thiol) to the gold prior to MUTEg passivation (and thus included in the SPR measure) but becomes displaced upon MUTEg passivation (and thus is not present in subsequent protein binding assays).

The employment of saline PBS buffer to perform the washing step has been reported in many previous works (Eisele et al. 2010, Schoch et al. 2012, Kapinos et al. 2014, Kapinos et al. 2017, Eisele et al. 2018) to safely release non-specifically bound molecules from the surface, while not interfering with the covalently bound molecules. Indeed, this is fully consistent with our observations: All FG-Nup layers are washed in PBS for several minutes after the FG-Nup incubation until a plateau in frequency is reached. This indicates that all non-specifically bound molecules are washed away before the next incubation step.

Strong physisorption of FG-Nups and subsequent misplacing by MUTEg are unlikely, since it is inconsistent with our new data. If a significant portion of Nups was physisorbed and afterwards displaced by MUTEg, then we should have found that the amount of removed Nups is dependent on their incubation concentration, especially so given the large excess of MUTEg employed in the experiment. However, we observe that the MUTEg passivation consistently yields very similar frequency and dissipation shifts, both when performed on pre-bound NupX brushes (100nM, 1µM, or 2µM) as well as on bare gold.

Also, I notice that the surface was washed in pure water and ethanol prior to drying. Did the authors check if this affects the amount of bound FG Nups?

The short rinsing in pure water and ethanol was performed to remove salts and help drying the sample. To confirm that such rinsing steps did not affect the amount of bound mass, we employed QCM-D and flushed MilliQ and 50% EtOH on gold surfaces that had been coated with NupX at 100nM, 1µM, and 2µM. We included such additional controls now in the SI in Fig. S8. The data indicate that no FG-Nups are removed from the surface in these two steps.

Minor comments:

1. Page 2: "As is evident from the multitude of transport models, no consensus on the NPC transport mechanisms has yet been reached." – the multiplicity of models *per se* is not an indicator of a lack

of consensus (or more generally, of the state of this research area), in particular considering that the references provided cover almost 2 decades of research.

We rephrased text accordingly on p. 2

2. Page 6: The authors use a two-component Langmuir isotherm to quantify binding strength. The authors should clarify the meaning of the obtained values to avoid misleading interpretation by lay readers. Considering the characteristics of the interacting molecules (*i.e.*, their multivalency), an argument can be made that it is unlikely the two K_d values represent two well-defined yet distinct types of binding sites within the FG nup films. Rather, the obtained values should be considered 'effective' and may represent a wide (and Kap95 concentration dependent) spectrum of binding sites and strengths. This was already discussed in previous work by others, and should be acknowledged.

We agree. We rephrased the text accordingly on p. 6, justifying the choice and the meaning of the two-component Langmuir isotherm to fit our QCM-D data.

Independent from the above, yet related to it, I note that the authors implicitly assume that frequency shifts are proportional to adsorbed amounts of Kap95. More often than not, this is not the case (for examples, see Reviakine et al Anal Chem 2011, 83:8838, and references therein). This is another reason that the determined K_d values should be considered effective values, and the authors should acknowledge this.

We thank the reviewer for the suggestion. We made this clearer now in the text on p.17. We would like to point out that in this case it is fully justified to assumed a linear correlation between frequency shift and Kap absorption since, unlike for the Nup layers, the D/f ratio was always very low ($< 10^{-8} \text{ Hz}^{-1}$, Fig. S10), motivating the use of the Sauerbrey relation here.

3. The authors should point out that NupX is substantially shorter than any of the native GLFG domains.

We clarified this point in the section on our design method on p. 4.

4. Page 7: The detailed description of the FG nup density profile refers to a 4.0 nm anchor spacing. This could be made cleared in the text.

We clarified this in the caption of Fig. 2 and the reference to Fig. 2 on p. 7.

5. Page 7: Kap95 was modelled as a sphere matching the total charge density of the protein. Is this meaningful, *i.e.* are all charged amino acids in Kap95 exposed on the protein surface?

We believe this modelling choice to be valid since virtually all charged residues in Kap95 are exposed on the surface of the protein. However, we have now corrected the statement to reflect the actual implementation of the charged residues in Kap95 where we distributed the *net* charge of the protein over the surface of the particle. We have rephrased the text on p. 20.

6. Page 8: The authors attribute transient dips in current to single molecule translocation. This merits further comment. Do the authors propose protein translocation to be 'single file'? This would be surprising given that an NPC is thought to contain many nuclear transport receptors at any given time (as the authors acknowledge in the introduction). Based on their affinity measurements for NupX and literature data for other FG nups, the authors should be able to estimate the equilibrium amount of Kap95 expected to be located within the pore. How does the result compare with single molecule translocation?

We thank the reviewer to bring up this interesting point. As the reviewer points out, transient dips in the current are indeed attributed to 'single file' translocations, while given the K_d measured for Kap95 and NupX brushes, one would expect to also have a certain amount of Kaps simultaneously present in the pore at any given time. While we did sometimes measure a slight decrease of the current baseline upon flushing Kap95, potentially associated with more Kap95 in the pore, we did not observe any consistent evidence that was worth reporting on in this work. However, we are planning to characterize this more systematically in an extensive follow-up study involving both synthetic and native FG-Nups and higher Kap95 concentrations.

7. Page 9: Can the authors comment on why the blockade times in NupX functionalised pores are longer for BSA than for Kap95?

We attribute this to the fact that BSA affinity to NupX is very low (Fig. 2i) as compared to Kap95. The transport times of diffusion through NPCs (and their mimics) can be described using first-order kinetics where the transport time depends on the free energy barriers associated with permeation through the FG-Nup meshwork (see for example Timney et al., JCB 2016, where the molecular mass, free energy barrier and experimental transport times are coupled to each other). The decreased energy barrier experienced by Kap95 due to transient binding to the FG-repeats of NupX underlies its selective transport and explains its reduced dwelling time as compared to BSA. We added a comment on p. 9 clarifying this.

8. Page 10: "Note that we adopted a critical protein density of 85 mg/ml from the earlier work on Nsp131 while employing a different dependency of the conductivity on the local protein density (Materials and Methods)." The implications are difficult to understand in the context of the main section. Can the authors clarify, in a simple way, how the seemingly excellent match between experiments and simulations shown in Fig. 4f should be interpreted?

We have phrased this more clearly now, and elaborated on the significance of the correspondence between the experimental conductance and the values calculated from our simulations on p.11 of the revised manuscript.

9. Page 11: The authors provide an estimate of a binding energy of just 3 $k_B T$. This appears rather low compared to the effective affinities determined experimentally on planar NupX films. Can the authors comment on this discrepancy?

The referee is correct in pointing out that the Kap95 binding energy from our nanopore simulations differs notably from the experimentally obtained apparent binding affinities in brushes. We believe

that this difference in binding energy is a consequence of the differences in geometry and the orientation of the reaction coordinate, a point which we have now clarified on p.12.

Briefly: One would not expect the same binding free energies for planar and pore geometries. The reaction coordinate in our nanopore simulations illustrates the sideways entry of NTRs into a film of Nups that decorates the pore interior, which differs notably from the orientation of the reaction coordinate that describes how NTRs impinge on a Nup-coated surface. Note also that the strong binding observed in the brush experiments correspond to a slow off-rate, and such binding affinities would in a nanopore setting yield dwell times that are incompatible with the fast transport times of NTRs observed in both *in vitro* experiments shown in our work and *in vivo* experiments on nuclear transport. Finally, the energy well of several $k_B T$ that Kap95 particles experience in our simulations is consistent with earlier simulation work on NPCs and NPC-mimics from various groups, which we now cite more explicitly on p.12.

10. Page 16: The meaning of 'increase in viscoelasticity' is not clear.

We rephrased the text on p.16.

11. Page 17: The authors stated QCM-D data were filtered. This is unusual; please clarify what type of treatments the original data were subjected to.

We initially used a smoothing function for some traces to remove the high-frequency noise, for display purposes. Importantly, this did not alter the observed trends in any way. We now replotted the new QCM-D traces without using any type of smoothing, and removed the word 'filtering' from the text on p.17.

12. Fig. 1b, what does the pink colour for amino acids represent?

This refers to the amino acids belonging to the extended domain (that comprises a high C/H ratio domain), and is used consistently throughout the work. We now made the colouring choices more explicit in the caption of Fig. 1b and clarified the reference to the figure and colouring scheme on p.4.

13. Fig. 2: Details in the dissipation response are difficult to see. Can the authors adjust the scale to facilitate a clearer view of the data? Ditto for all QCM-D data shown in the manuscript and supporting information.

We adjusted all scales accordingly.

14. Fig. 4b, g, h: The densities appear asymmetric along the axis. Can the authors comment on this?

The same number of proteins is anchored on either side of the $z=0$ axis. However, since we employ a triangulated lattice with an even number of rows, the anchoring pattern is not entirely symmetric in the $z=0$ plane. This induces a slight asymmetry (differences of $\sim 10\%$ between the top and the bottom part of the central lobes, resp.) in the time-averaged protein density distributions. This effect is also present in earlier work with a similar simulation setup, see Ananth et al. eLife 2019. This does not affect the conclusions of our simulations: the density profiles in Fig. 4c,d are averaged over the entire pore height and the conductance calculation (Fig. 4f) comprises an integration over the volume of the

nanopore for short time windows, and both quantities are insensitive to the slight variation of the density near the centre of the pore. Moreover, we do not observe any clear preference of the model NTR particles to localize on one specific side in Fig. 5e, again indicating that there are no notable consequences of the slight asymmetry. We have added this clarification now to the main text on p.10.

15. Supplementary figures are not numbered by order of appearance in the main text. Moreover, some supplementary figures are not even mentioned in the main text (*e.g.*, Fig. S7). This makes reading unnecessarily cumbersome.

We thank the reviewer for the suggestion. We renumbered figures accordingly. Since we grouped the SI figures per technique (in the same order as they appear in the main text), a single exception in the ordering occurs.

16. Fig. S1: Can the authors also show a gel demonstrating the purity of the Kap95 prep.

We added a gel for Kap95 prep in Fig. S1.

17. Fig. S2: The authors claim the frequency increase upon washing is due to release of proteins. Another common source of rapid shift upon washing is differences in the solution viscosity or density. How did the authors discriminate between these two scenarios?

The Kap95 solutions consisted of PBS which was as well used to rinse off the proteins. The very low concentrations of proteins are not expected to affect the viscosity of the solution. A few reported examples can show that concretely: a 2.5 μ M solution of BSA (that was also diluted in PBS) did not cause any steep change in the frequency response (Fig. 2i); a 500nM solution of Kap95 flushed onto a Nsp1-S coated surface did not cause any change, neither in frequency nor in dissipation (Fig. 2h); If the μ M concentrations of Kaps or BSA were to affect the buffer viscosity to a non-negligible extent, then both the frequency and dissipation responses would have showed this clearly throughout all binding experiments on QCM-D. However, that was not the case. Instead, we do attribute such increase in frequency due to the washing away weakly bound Kaps.

18. Fig. S3: The authors claim the frequency shift for MUTEG binding is inversely proportional to NupX binding. The authors will want to demonstrate this is (quantitatively) true, or revise the claim for clarity.

We have removed this phrasing from the SI since, based on our new measurements, we do not find a strong dependence between the MUTEG binding and the pre-bound NupX.

19. Fig. S7: This plot is potentially very informative, but essential information is missing to assess this in full. Are the data shown for FG Nup incubation alone, or do they include the MUTEG passivation step? The latter would be more informative because this reflects the properties of the film used for protein binding studies. Also, I am surprised the ratios of dissipation shifts over frequency shifts are positive. They should be negative.

We corrected and clarified Fig. S7 (now Fig. S9) as suggested by the reviewer. Given that the new data show a very small change in dissipation upon MUTEG coating, while yielding similar shifts in frequency, we do still find it more interesting to plot the D/f shifts for the FG-Nup films alone.

Reviewer #4 (Remarks to the Author):

See separate file.

Review of 3rd revised version

The authors provide compelling evidence that they have identified and resolved problems with their MUTEG compound. The extensive new QCM-D data presented in the revised manuscript addresses MAJOR point 1 in the previous review in a satisfactory manner. It is also reassuring that the previous conclusions with regard to Kap95 and BSA binding remain essentially unaffected.

Unfortunately, concerns persist regarding the correlation of experimental data and predictions obtained from simulations. I recall that this is a point I had raised since the first version of the manuscript.

The authors report that the NupX film thickness as determined experimentally by the Sauerbrey equation is roughly 50% of the thickness obtained from the simulation data. An argument is then made that the values are consistent considering the approximative nature of the Sauerbrey equation. I agree with the authors that the Sauerbrey equation underestimates the film thickness for soft films. For sufficiently rigid films, however, the Sauerbrey equation is expected to provide a good approximation of the film thickness. In the case of the NupX film, the $\Delta D/-\Delta f$ ratio is rather low (i.e. $\Delta D/-\Delta f \ll 0.4 \times 10^{-6} \text{ Hz}^{-1}$, indicative of a rather rigid film) and one can expect the Sauerbrey equation to provide a fairly good approximation of the film thickness.

The table below provides a comparison of results, where I have included data from the authors' manuscript (Fig. 2a, for NupX), from one of the cited papers (ref. 45, for Nsp1), and from another paper (Eisele et al *Biophys J* 2013, for a glycosylated construct of Nup98). The latter is relevant because the $\Delta D/-\Delta f$ ratio for Nup98-glyco is comparable to NupX ($0.045 \times 10^{-6} \text{ Hz}^{-1}$). The table includes film thickness data obtained using the Sauerbrey equation ($d_{\text{Sauerbrey}}$). It also includes data obtained by viscoelastic modelling ($d_{\text{viscoelastic modelling}}$, taken from the publications) which properly accounts for film softness. It can be seen that both thickness values are almost identical for Nup98-glyco. Even for Nsp1, which is substantially softer than NupX and Nup98-glyco ($\Delta D/-\Delta f = 0.082 \times 10^{-6} \text{ Hz}^{-1}$) the thickness underestimation by Sauerbrey remains moderate (20%) compared to the above-mentioned 50% discrepancy.

FG nup film	NupX	Nsp1	Nup98-glyco
$\Delta f_5/5$ (Hz)	-39	-142	-120
ΔD_5 (10^{-6})	1.8	11.6	5.3
$\Delta D_5/(-\Delta f_5/5)$ (10^{-6} Hz^{-1})	0.046	0.082	0.044
$d_{\text{Sauerbrey}}$ @ $p = 1.05 \text{ g/cm}^2$ (nm)	6.7	24.3	20.6
$d_{\text{Viscoelastic model}}$ @ $p = 1.05 \text{ g/cm}^2$ (nm)		$30.5^{+4.5}_{-1.5}$	20 ± 3
$d_{\text{Sauerbrey}} / d_{\text{Viscoelastic model}}$		$0.80^{+0.04}_{-0.11}$	$0.97^{+0.24}_{-0.07}$

The above analysis suggests that there is a discrepancy of 50% in the film thickness. This is a substantial difference, and of concern because major conclusions of the manuscript (e.g. explanation of the pore conductance vs pore diameter curve and the Kap95 translocation rates) are based on quantitative correlation between experiments and simulations. A case in point: translated to the pore geometry, one would expect that a central channel remains open at much smaller pore diameters than predicted by the simulations. The authors need to address the inconsistency

between the seemingly excellent correlation of the data in Fig. 4f (pore geometry) and the apparent discrepancy in film thickness (planar geometry). From the presented data it is difficult to assess if the issue reflects limitations of the simulations, the control of experimental parameters, or both.

The majority of other points raised in my previous review has been addressed satisfactorily. Some minor points persist, however, as list below.

- In response to the authors' reply to MAJOR comment 4 in the previous revision. I concur with the authors that MUTEG is unlikely to affect the FG Nups that are properly grafted to the gold. However, have the authors verified that all adsorbed FG Nups are indeed attached via a cysteine to the gold? Physisorption to gold is common for proteins and it is unclear if this is also happening here in parallel to the cysteine-mediated binding. The washing data in Fig S2 indeed indicate physisorption take place, and demonstrates some of the FG Nups can be readily released. However, it remains unclear if physisorbed material is still retained after washing. Possible controls are to test the response of the FG Nup films to a denaturing agent that reduces the physical interactions with the gold and other surface-bound FG nups, and/or to test the binding and unbinding of FG Nups that lack a cysteine.
- In response to the authors' reply to MINOR comment 3 in the previous revision. It is not clear why the authors invoke the $\Delta D / -\Delta f$ ratio here. The non-linear relationship between frequency shifts and adsorbed amounts is due to coverage-dependent changes in the fraction of hydrodynamically trapped solvent. This effect is known to occur even for rigid films, and I refer to previously mentioned references for examples.
- In response to the authors' reply to MINOR comment 9 in the previous revision. It is remarkable that the simulations predict a 7-fold increase in the free energy of Kap95 binding when moving from the planar to the pore geometry (whilst maintaining quite similar local FG nup densities). The authors propose that the direction of entry into the brush matters. Have they tried to corroborate this? Given the flexibility of FG nups, and their moderate stretching normal to the grafting surface, I would have expected the chains to present very little preferred orientation at a length scale commensurate with the size of Kap95.
- On page 6, the authors state that a 'clear concentration-dependent equilibrium amount of Kap95 molecules bound to the NupX brush'. The data in Fig. 2b do not appear to support this claim. Clear plateaus are lacking at virtually all Kap95 titration steps. This needs to be considered in the interpretation of the K_d values. I also notice that the vast majority of Kap95 remains irreversibly bound to the NupX film, suggesting a very low off rate. Is this consistent with the order of magnitude of the affinities obtained by the two-component Langmuir-isotherm fit (producing K_d values in the 100s of nanomolar to micromolar range, although it was not clear how large each of the two fractions is)?
- Fig. S3B: The panel indicates data for gold with MUTEG is shown, but the legend indicates gold with mPEG is shown. Could it be clarified that this is the same molecule?
- Fig. S2: Can more data prior to NupX incubation be shown to demonstrate the QCM-D responses (i.e. baselines) were stable? 10 to 20 min would be ideal. Also, can the authors comment on the extended lag time between NupX incubation and binding in panel C, which is unexpected.

Detailed point-by-point reply to the latest (4th) report of reviewer 4

The authors provide compelling evidence that they have identified and resolved problems with their MUTEG compound. The extensive new QCM-D data presented in the revised manuscript addresses MAJOR point 1 in the previous review in a satisfactory manner. It is also reassuring that the previous conclusions with regard to Kap95 and BSA binding remain essentially unaffected.

We thank the reviewer for acknowledging that all irregularities in our QCM-D data have been resolved.

Unfortunately, concerns persist regarding the correlation of experimental data and predictions obtained from simulations. I recall that this is a point I had raised since the first version of the manuscript.

In the initial version of the manuscript, the reviewer rightly pointed out that obtaining an accurate estimate of the NupX grafting distance (that subsequently is used as input for our simulations) directly from the QCM-D is not possible given the hydrated nature of such brushes. Due to those concerns, we initiated a collaboration with the Dahlin lab, well-known for their expertise in the SPR technique, to quantify the dry mass density. Based on those new estimates, we repeated all experiments on QCM-D using the same coating conditions (protein concentrations, incubation time, buffers) as in the SPR measurements. In this way we confirmed that NupX-brushes can be close-packed and that the simulations were indeed carried out for a relevant range of grafting distances, thus resolving the initial concerns by the reviewer.

Although all these new measurements are judged as solid by the reviewer, *yet new* concerns are now raised about the thickness of the NupX layer extrapolated from QCM-D data, and the one from simulation results. This is a different issue, which we address in the next paragraph.

The authors report that the NupX film thickness as determined experimentally by the Sauerbrey equation is roughly 50% of the thickness obtained from the simulation data. An argument is then made that the values are consistent considering the approximative nature of the Sauerbrey equation. I agree with the authors that the Sauerbrey equation underestimates the film thickness for soft films. For sufficiently rigid films, however, the Sauerbrey equation is expected to provide a good approximation of the film thickness. In the case of the NupX film, the $\Delta D/\Delta f$ ratio is rather low (i.e. $\Delta D/\Delta f \ll 0.4 \times 10^{-6} \text{ Hz}^{-1}$, indicative of a rather rigid film) and one can expect the Sauerbrey equation to provide a fairly good approximation of the film thickness.

The table below provides a comparison of results, where I have included data from the authors' manuscript (Fig. 2a, for NupX), from one of the cited papers (ref. 45, for Nsp1), and from another paper (Eisele et al *Biophys J* 2013, for a glycosylated construct of Nup98). The latter is relevant because the $\Delta D/\Delta f$ ratio for Nup98-glyco is comparable to NupX ($\sim 0.045 \times 10^{-6}$

Hz⁻¹). The table includes film thickness data obtained using the Sauerbrey equation ($d_{\text{Sauerbrey}}$). It also includes data obtained by viscoelastic modelling ($d_{\text{viscoelastic modelling}}$, taken from the publications) which properly accounts for film softness. It can be seen that both thickness values are almost identical for Nup98-glyco. Even for Nsp1, which is substantially softer than NupX and Nup98-glyco ($\Delta D / -\Delta f = 0.082 \times 10^{-6} \text{ Hz}^{-1}$) the thickness underestimation by Sauerbrey remains moderate (20%) compared to the above-mentioned 50% discrepancy.

FG nup film	NupX	Nsp1	Nup98-glyco
$\Delta f_5 / 5$ (Hz)	-39	-142	-120
ΔD_5 (10^{-6})	1.8	11.6	5.3
$\Delta D_5 / (-\Delta f_5 / 5)$ (10^{-6} Hz^{-1})	0.046	0.082	0.044
$d_{\text{Sauerbrey}}$ @ $\rho = 1.05 \text{ g/cm}^2$ (nm)	6.7	24.3	20.6
$d_{\text{Viscoelastic model}}$ @ $\rho = 1.05 \text{ g/cm}^2$ (nm)		$30.5^{+4.5}_{-1.5}$	20 ± 3
$d_{\text{Sauerbrey}} / d_{\text{Viscoelastic model}}$		$0.80^{+0.04}_{-0.11}$	$0.97^{+0.24}_{-0.07}$

The above analysis suggests that there is a discrepancy of 50% in the film thickness. This is a substantial difference, and of concern because major conclusions of the manuscript (e.g. explanation of the pore conductance vs pore diameter curve and the Kap95 translocation rates) are based on quantitative correlation between experiments and simulations. A case in point: translated to the pore geometry, one would expect that a central channel remains open at much smaller pore diameters than predicted by the simulations. The authors need to address the inconsistency between the seemingly excellent correlation of the data in Fig. 4f (pore geometry) and the apparent discrepancy in film thickness (planar geometry). From the presented data it is difficult to assess if the issue reflects limitations of the simulations, the control of experimental parameters, or both.

We do *not* agree with the viewpoint of the reviewer that the lack of an exact numerical match between the NupX brush heights predicted by simulations at two grafting distances, and the estimate based on the Sauerbrey equation applied to the data of Fig. 2a, invalidates the main conclusions of this study.

To re-iterate the goal of our study: we successfully demonstrated the realization of nuclear pore complex mimics that rely on an artificial FG-Nucleoporin, which we designed using a rational set of design rules. Our main findings, namely selective transport of NTRs over inert proteins through NupX-functionalized nanopores and the selective interaction of NupX towards NTRs over inert proteins are independently supported by two types of experiments, and two types of simulations. The validity of these findings does not depend on an exact numerical correspondence between observables from the experiments or simulations, which all have their own constraints. We note that the reviewer does not provide any critique on the validity of our QCM-D/SPR-measurements in the current version of the manuscript, nor did any of the other reviewers in earlier versions. Likewise, none of the other reviewers presented critiques on the computational model. We are confident of the performance of our model, which has been used before to successfully study protein distributions and selective transport

in NPCs and NPC mimics (by us and other labs), has correctly reproduced brush heights measured by SPR of Nup62 brushes (Ghavami et al. BPJ 2015) and has been successfully used in similar combined experimental and computational studies (Ananth et al. eLife 2019).

To illustrate why the main findings are not impacted by a numerical difference between estimates of one observable, we refer to the two examples presented by the reviewer: 1. The ionic conductance relation in Fig. 4f relies on a zero-parameter fit, with the unknown dependence of conductivity on protein density specifically chosen for NupX (as indicated in the materials section). We refrained from implying that this result on ionic transport is relevant to particle transport selectivity, given that it does not play a causal role. 2. In a second example, the referee indicates that we quantitatively correlate results from experiments and simulations on translocation rates of Kap95. This is incorrect: at no point do we draw a quantitative correlation between our nanopore simulations and the translocation experiments in Figs. 3 and S15. If anything, we only noted that the formation of a hole seemed consistent with the increase in transport rates for large diameter pores.

Furthermore, the height comparison proposed by the reviewer is flawed. The comparison between the experimental estimate of the layer height and the simulated brush height is difficult to make due to several complicating factors. Firstly, the exact grafting distance used in our Sauerbrey estimate (based on incubation conditions of 1 μ M, Fig. 2a) is unknown, which precludes a direct one-to-one comparison. Secondly, although we followed a literature example in estimating the height of our simulated brushes from the density profiles, there is not a clear guideline for where a brush exactly 'ends' and hence what defines the edge and thus precise height of the brush. Lastly, we re-iterate that the experimental estimate is a lower limit only (as extensively discussed in previous rebuttals). When accounting for uncertainties in the viscoelastic modeling (note that the reviewer accidentally inverted the division in calculating $d_{\text{sauerbrey}}/d_{\text{viscoelastic}}$ for Nup98) one arrives at the conclusion that even the 'stiffest' example of an FG-Nup presented by the reviewer can be off by at least $\sim 10\%$. The thickness underestimation by the Sauerbrey equation (which can still be significant), together with the other uncertainties in the brush height estimates, could very well explain why notable quantitative differences can arise in this comparison.

We added a note on page 7 to clarify the above point.

The majority of other points raised in my previous review has been addressed satisfactorily.

We thank the reviewer for this positive point of appraisal.

Some minor points persist, however, as list below.

1) In response to the authors' reply to MAJOR comment 4 in the previous revision. I concur with the authors that MUTEG is unlikely to affect the FG Nups that are properly grafted to the gold. However, have the authors verified that all adsorbed FG Nups are indeed attached via a cysteine to the gold? Physisorption to gold is common for proteins and it is unclear if this is also happening here in parallel to the cysteine-mediated binding. The washing data in Fig S2 indeed indicate physisorption take place, and demonstrates some of the FG Nups can be

readily released. However, it remains unclear if physisorbed material is still retained after washing. Possible controls are to test the response of the FG Nup films to a denaturing agent that reduces the physical interactions with the gold and other surface-bound FG nups, and/or to test the binding and unbinding of FG Nups that lack a cysteine.

We do not see the added benefit of adding further denaturants in this case since our FG-Nup layers are stable under washes using 0.2M NaOH (Fig. S6 shows full removal of Kap95 but no additional mass is removed), milliQ, and 50%EtOH (Fig. S8). These results all indicate that our FG-Nups that are left after the first PBS wash are covalently bound and not simply physisorbed.

2) In response to the authors' reply to MINOR comment 3 in the previous revision. It is not clear why the authors invoke the $\Delta D/\Delta f$ ratio here. The non-linear relationship between frequency shifts and adsorbed amounts is due to coverage-dependent changes in the fraction of hydrodynamically trapped solvent. This effect is known to occur even for rigid films, and I refer to previously mentioned references for examples.

The measured $\Delta D/\Delta f$ upon Kap95 absorption was very low ($< 0.01 \times 10^{-6} \text{ Hz}^{-1}$) which motivated the use of the Sauerbrey relation to estimate the amount of adsorbed protein. The referee is right in stating that non-linear processes may occur thereby altering the estimation of the dissociation constants. We added this to our comments on the apparent nature of the dissociation constants on p.6.

3) In response to the authors' reply to MINOR comment 9 in the previous revision. It is remarkable that the simulations predict a 7-fold increase in the free energy of Kap95 binding when moving from the planar to the pore geometry (whilst maintaining quite similar local FG nup densities). The authors propose that the direction of entry into the brush matters. Have they tried to corroborate this? Given the flexibility of FG nups, and their moderate stretching normal to the grafting surface, I would have expected the chains to present very little preferred orientation at a length scale commensurate with the size of Kap95.

Perhaps we were not clear enough in describing the concept of the reaction coordinate. We do not base our explanation on a preferred orientation of the FG-Nup towards the NTR. Rather, the PMF describes the physicochemical environment that the NTR senses along a chosen path (the reaction coordinate, in this case, the z-direction). The reaction coordinate that describes traversing through the channel with FG-Nups laterally attached at the pore surface is fundamentally different from the one describing perpendicularly entering a brush layer of FG-Nups.

In drawing the comparison between the pore and the planar geometry, we should consider both the dense (Fig. 2I) and the sparser case (Fig. S11): an energy well of approx. -50 kJ/mol, found in the dense brush, corresponding to a nanomolar effective binding affinity, and a free energy difference in the sparser brush of approximately -30 kJ/mol, corresponding to a micromolar binding affinity. Whereas the former deviates, the latter value is very comparable to *in vitro* values from various SPR and QCM-D experiments reported in the literature (see for

example Wagner et al., BPJ 2015, Kapinos et al. J. Cell. Biol. 2017). We now included these estimates on p.8 of the manuscript.

Once more we would like to emphasize that other authors in the scientific literature (e.g., Tetenbaum-Novatt et al., Mol. Cell. Prot. 2012) have already pointed out that the high (micromolar to tens of nanomolar) binding affinities (and accompanying free energies) found in surface binding experiments on FG-Nups and NTRs should not be expected in the pore geometry since that would preclude the fast ~ms transport observed in NPCs and NPC mimics. Our findings for both geometries are thus consistent.

4) On page 6, the authors state that a 'clear concentration-dependent equilibrium amount of Kap95 molecules bound to the NupX brush'. The data in Fig. 2b do not appear to support this claim. Clear plateaus are lacking at virtually all Kap95 titration steps. This needs to be considered in the interpretation of the K_d values. I also notice that the vast majority of Kap95 remains irreversibly bound to the NupX film, suggesting a very low off rate. Is this consistent with the order of magnitude of the affinities obtained by the two-component Langmuir-isotherm fit (producing K_d values in the 100s of nanomolar to micromolar range, although it was not clear how large each of the two fractions is)?

Although saturation is not fully achieved for a few concentration steps (due to the extremely slow binding of Kap95 to NupX), there is a clear Kap concentration-dependence of the frequency response. Similar behaviour has been reported also in previous works (Kapinos et al 2014, and 2017) for Kap-beta1 (human homolog of the yeast Kap95) association to native FG-Nups, where full saturation could not be achieved with similar incubation times. Such slow binding rates may be caused by mass transport limitations that are inherent to the technique and could ultimately alter the estimation of the binding affinity values, as we stated now on p.6.

Concerning the dissociation rate, we did reproducibly find that Kaps bind strongly to the NupX layer leading to a remarkably low off-rate. Such low off-rates were also observed for dissociation of Kap-beta1 from native FG-Nups (Kapinos et al 2014, and 2017) and further investigated by Hayama et al. 2019, who attribute it to the influence of mass transport limitations that are inherent to such surface techniques like QCM-D or SPR. We clarified this point now on p. 6.

5) Fig. S3B: The panel indicates data for gold with MUTEG is shown, but the legend indicates gold with mPEG is shown. Could it be clarified that this is the same molecule?

We clarified this now in the caption of Fig. S3B.

Fig. S2: Can more data prior to NupX incubation be shown to demonstrate the QCM-D responses (i.e. baselines) were stable? 10 to 20 min would be ideal. Also, can the authors comment on the extended lag time between NupX incubation and binding in panel C, which is unexpected.

Upon the reviewer's request we now added more minutes to the beginning of the QCM-D trace in Fig. S2.

We do not find the lag time for the NupX incubation and binding in panel C unexpected: while the flow-rate is constant for all sensors, the length of the tubings prior the chip chamber can slightly vary and this will induce different levels of mixing/dilution at the interface between the NupX solution and the preceding running buffer (PBS), which can show up as an apparently slower or faster initial binding. Besides that, we do find the binding rate in panel C to be comparable to the one in panel B. Since we are interested in the final frequency shifts at equilibrium after the buffer wash, the initial binding kinetics are not relevant for the current study.

Reviewer #5 (Remarks to the Author):

It was my pleasure to review the manuscript by Dekker et al. on “A designer FG-Nup that reconstitutes the selective transport barrier of the Nuclear Pore Complex”. The manuscript is very well-written, with interesting novel results corresponding to its title. To summarize my opinion from the outset, I think both the novelty and the quality of the work makes it suitable for Nat Comm. That said, I was asked specifically to comment on the dispute with previous Reviewer 4. Technically, I think Reviewer 4 makes valid points, to which I would also like to add some. However, like the authors, I do not share the opinion that these deviations between the experimental characterization of the system and the simulations have a major impact on the work's central conclusions. I basically share the authors' opinion that these concerns were taken care of in the current form of the manuscript.

Suppose the authors acknowledge the difference and stick to the qualitative agreement (and within a factor of 2) in support of their model, for which they also can add some of the arguments below, and (in my preference) rewrite or remove the experimental determination of K_d (see below). In that case, I think this manuscript should be published in Nat Comm almost as it is.

Detailed points

The argument regarding the deviation between the Sauerbrey thickness from QCM-D and the in silico modeled thickness is too large to explain in the way that the authors do it. To me, the argument is slightly circular. Indeed, the low dissipation and DD/D_f ratio indicate that the deviation should be small based on lots of published works. This makes it reasonable to use it, despite that the film is clearly not homogeneous in peptide density and containing a high amount of water. Although the authors' point that “no one knows where a brush ends” is valid, it is not reasonable from this data to argue that half the brush would be so dilute that it does not register in QCM when the technique is known to be most sensitive to dilute extended soft polymers. This argument is also not consistent with the segment density profile from the simulations. However, this quantitative discrepancy is not uncommon. I think the main point is that the SPR/QCM-D and the simulations independently show that a brush of disordered Nupx is formed.

If I understand it right, the authors claim that the grafting density to determine the density for the Sauerbrey relation calculation is not known and that this could explain the deviation. At the same time, they argue the equivalence of the SPR and QCM measurements, which means that the density is known and can be used for the Sauerbrey calculation or even to model the data iteratively. In any case, based on the numbers in the manuscript, this is a 5% discrepancy as best and cannot explain the difference between modeling and QCM thickness.

However, except for one sentence (which I agree with Reviewer 4 goes too far) that says that there is a good (quantitative) agreement between modeling and QCM experiments, I don't think this discrepancy impacts the manuscript. They establish that quite dense and, for practical purposes, homogeneous brushes form from the Nupx. In the central experiments on the nanopores, the geometry has been changed. A concave brush is different in its structure from a planar brush. There are well-known theoretical and simulation works that could be used by the authors here. They do not show the same segment distribution, and they will also extend farther compared to brushes with equivalent grafting density and molecular weight. Experimentally, a larger fraction of the pore diameter is, therefore, expected to be blocked than the planar brush experiments will predict. This difference changes non-linearly with pore diameter and becomes larger for small pores. This curvature effect is relevant for the small pores. The agreement between the model and the data for the nanopore experiments is convincing.

However, I think that the authors should acknowledge that the grafting density in the nanopores might deviate significantly from what they observe on a planar surface. There is also literature

indicating macromolecular adsorption in crowded spaces, such as pores, is affected by the geometric and steric constraints. This might be the case here as well, but the authors assume that the numbers found from the planar brush experiments will translate to the pores. I would assume that these are an upper limit to the grafting density, an upper bound to brush extension in the convex parts at the openings, and a lower bound to the brush thickness in the pore's concave inner parts.

More puzzling is the choice of techniques and models applied to measure the dissociation constant of the Kap95 interaction with the Nup brushes. However, here the authors clearly acknowledge the discrepancy between those results, the observed translocation rate, and the modeled values. Thus, it does not substantially influence the conclusions, and I mainly wonder why these measurements were included in the manuscript performed in this way, and when they don't contribute to our understanding of the results.

Firstly, why was QCM used to determine the dissociation for an interaction that can occur inside the brush as well as on it? QCM is not a suitable technique for determining K_d despite that it is often used. This is primarily because, as the authors acknowledge in other parts of the manuscript, that the water coupling per molecule is unknown. Thus, the response is non-linear to concentration. This problem becomes worse when we, in this case, can consider that the adsorbing Kap95 also can replace already trapped water by binding into the brush. The authors used SPR to study Nup grafting. Why they did not use SPR, which is a suitable technique for these measurements, also for the K_d measurements is a riddle to me.

Adding to the problem of the K_d measurements is that the authors say that they use what they call the Langmuir model with two rate constants. That might be a bit of a misnomer, but letting that slide, this model for determining K_d still requires a dynamic equilibrium, i.e., that the interaction is reversible. Else, the dissociation constant cannot be determined in this way. The data shows with all clarity that the Kap95-Nup interaction performed with QCM on the NUP-brushes is not reversible and out of equilibrium. Thus, this way of calculating K_d is simply incorrect.

Finally, a few common errors such as writing homogenous (same genes) instead of homogeneous and absorb instead of adsorb were spotted in the manuscript.

Signed,

Erik Reimhult

Reply to Referee #5

It was my pleasure to review the manuscript by Dekker et al. on “A designer FG-Nup that reconstitutes the selective transport barrier of the Nuclear Pore Complex”. The manuscript is very well-written, with interesting novel results corresponding to its title. To summarize my opinion from the outset, I think both the novelty and the quality of the work makes it suitable for Nat Comm.

We thank the reviewer for the positive appraisal and for recommending the publication of the paper in Nature Comm.

That said, I was asked specifically to comment on the dispute with previous Reviewer 4. Technically, I think Reviewer 4 makes valid points, to which I would also like to add some. However, like the authors, I do not share the opinion that these deviations between the experimental characterization of the system and the simulations have a major impact on the work's central conclusions. I basically share the authors' opinion that these concerns were taken care of in the current form of the manuscript.

We appreciate that the referee shares our view on the previous referee report.

Suppose the authors acknowledge the difference and stick to the qualitative agreement (and within a factor of 2) in support of their model, for which they also can add some of the arguments below, and (in my preference) rewrite or remove the experimental determination of K_d (see below). In that case, I think this manuscript should be published in Nat Comm almost as it is.

We thank the reviewer for the constructive points. We discuss these points individually in the sections below.

Detailed points:

1) The argument regarding the deviation between the Sauerbrey thickness from QCM-D and the in silico modeled thickness is too large to explain in the way that the authors do it. To me, the argument is slightly circular. Indeed, the low dissipation and DD/D_f ratio indicate that the deviation should be small based on lots of published works. This makes it reasonable to use it, despite that the film is clearly not homogeneous in peptide density and containing a high amount of water. Although the authors' point that “no one knows where a brush ends” is valid, it is not reasonable from this data to argue that half the brush would be so dilute that it does not register in QCM when the technique is known to be most sensitive to dilute extended soft polymers. This argument is also not consistent with the segment density profile from the simulations. However, this quantitative discrepancy is not uncommon. I think the main point is that the SPR/QCM-D and the simulations independently show that a brush of disordered Nupx is formed. If I understand it right, the authors claim that the grafting density to determine the density for the Sauerbrey relation calculation is not known and that this could explain the deviation. At the same time, they argue the equivalence of the SPR and QCM measurements, which means that the density is known and can be used for the Sauerbrey calculation or even to model the data iteratively. In any case, based on the numbers in the manuscript, this is a 5% discrepancy as best and cannot explain the difference between modeling and QCM thickness. However, except for one sentence (which I agree with Reviewer 4 goes too far) that says that there is a good (quantitative) agreement between modeling and QCM experiments, I don't think this discrepancy impacts the manuscript.

We thank the reviewer for this clear assessment of the argumentation that we present in the manuscript. Following the reviewer's suggestion, we rephrased several parts of the section on NupX-brushes. Concretely, we now acknowledge the differences between the simulated and experimental results and mention the fact that this does not impact the main findings of this work on p.7. Moreover, we refrain from stating that there is a good quantitative agreement between modeling and (QCM) experiments throughout the manuscript.

2) They establish that quite dense and, for practical purposes, homogeneous brushes form from the Nupx. In the central experiments on the nanopores, the geometry has been changed. A concave brush is different in its structure from a planar brush. There are well-known theoretical and simulation works that could be used by the authors here. They do not show the same segment distribution, and they will also extend farther compared to brushes with equivalent grafting density and molecular weight. Experimentally, a larger fraction of the pore diameter is, therefore, expected to be blocked than the planar brush experiments will predict. This difference changes non-linearly with pore diameter and becomes larger for small pores. This curvature effect is relevant for the small pores. The agreement between the model and the data for the nanopore experiments is convincing.

However, I think that the authors should acknowledge that the grafting density in the nanopores might deviate significantly from what they observe on a planar surface. There is also literature indicating macromolecular adsorption in crowded spaces, such as pores, is affected by the geometric and steric constraints. This might be the case here as well, but the authors assume that the numbers found from the planar brush experiments will translate to the pores. I would assume that these are an upper limit to the grafting density, an upper bound to brush extension in the convex parts at the openings, and a lower bound to the brush thickness in the pore's concave inner parts.

We thank the reviewer for pointing us towards the effect of the concave nanopore surface on the organization of our NupX proteins in such geometries. We now mention this effect and how it relates to our experiments and simulations on p. 11. In addition, we added a statement on the limiting value of the grafting distance in our nanopores to p. 10 and p. 22, and more explicitly refer to our sensitivity analysis where we showed that a 10% increase in grafting distance does not affect our simulation results appreciably on p. 11.

3) More puzzling is the choice of techniques and models applied to measure the dissociation constant of the Kap95 interaction with the Nup brushes. However, here the authors clearly acknowledge the discrepancy between those results, the observed translocation rate, and the modeled values. Thus, it does not substantially influence the conclusions, and I mainly wonder why these measurements were included in the manuscript performed in this way, and when they don't contribute to our understanding of the results.

Firstly, why was QCM used to determine the dissociation for an interaction that can occur inside the brush as well as on it?

We are aware of the limitations of QCM-D but like to briefly mention our original intentions in using the technique:

- 1) To qualitatively prove that NupX brushes interact selectively with Kap95 over similarly-sized inert proteins such as BSA.
- 2) To provide an estimate of the apparent dissociation constant for the binding of Kap95 to NupX and show consistency with published results on native FG-Nups.

QCM is not a suitable technique for determining K_d despite that it is often used. This is primarily because, as the authors acknowledge in other parts of the manuscript, that the water coupling per molecule is unknown. Thus, the response is non-linear to concentration. This problem becomes worse when we, in this case, can consider that the adsorbing Kap95 also can replace already trapped water by binding into the brush. The authors used SPR to study Nup grafting. Why they did not use SPR, which is a suitable technique for these measurements, also for the K_d measurements is a riddle to me. Adding to the problem of the K_d measurements is that the authors say that they use what they call the Langmuir model with two rate constants. That might be a bit of a misnomer, but letting that slide, this model for determining K_d still requires a dynamic equilibrium, i.e., that the interaction is reversible. Else, the dissociation constant cannot be determined in this way. The data shows with all clarity that the Kap95-Nup interaction performed with QCM on the NUP-brushes is not reversible and out of equilibrium. Thus, this way of calculating K_d is simply incorrect.

We thank the reviewer for the comment. To accommodate this criticism, we have made several changes to our interpretation of the QCM-D data in the manuscript:

- We now refrain from explicitly calculating K_d -values from our QCM-D Kap95 titration data, i.e., we removed the fitted isotherms from Fig. 2 and our description of the K_d -values on p. 6 as well as other references to experimental binding affinities on p. 3-4. Moreover, we made accompanying changes in the nanopore simulations and discussion sections (p. 12-13) and in our interpretation of the simulated binding affinities (p. 12).
- We have carefully rephrased our interpretation of the QCM-D Kap95 titration data (now Fig. 2d-f) on p. 6. We note that the observation of NupX brushes selectively interacting with Kap95 over BSA holds regardless of the adsorption model used or the concentration-dependent hydrodynamic effects. We opted for retaining the Kap95/BSA QCM-D titration traces, since these data still provide strong qualitative support for the main message of the manuscript.

Finally, a few common errors such as writing homogenous (same genes) instead of homogeneous and absorb instead of adsorb were spotted in the manuscript.

We thank the reviewer for pointing out these typos. We have carefully checked the manuscript for any of the mentioned mistakes and corrected these accordingly.

Reviewer #5 (Remarks to the Author):

Dekker et al. have revised the previously submitted manuscript on all the points that I raised. In most cases, the problematic parts were removed, such as the calculations of the dissociation constants and the overinterpretation of the quantitative agreement between experimental and simulation data. As I suggested in my previous report, removing these parts does not detract from the important points the manuscript makes. On the contrary, focusing on the main message and removing the parts of the quantitative data that was calculated and/or compared dubiously strengthens the manuscript.

I fully appreciate that the paper's main goal, including the QCM-D measurements, was to show the selective binding of Kap95 over BSA and its influence on transport through the pores. I think it is important to keep the data as the authors do. They conclusively show the point the authors want to make without calculating exact brush heights or affinity constants.

After reading the revised manuscript, I have no additional points to raise and recommend that the manuscript is published in Nature Communications.

Signed, Erik Reimhult